# *Wolbachia* action in the sperm produces developmentally deferred chromosome segregation defects during the *Drosophila* mid-blastula transition

**Brandt Warecki[1]\*†, Simon William Abraham Titen[1,2]†, Mohammad Shahriyar Alam[1], Giovanni Vega[1], Nassim Lemseffer[1], Karen Hug[1], Jonathan S Minden[3], William Sullivan[1]\***

[1]Department of Molecular, Cell, and Developmental Biology, University of California, Santa Cruz, Santa Cruz, United States; [2]Department of Biology and Chemistry, California State University Monterey Bay, Seaside, United States; [3]Department of Biological Sciences, Carnegie Mellon University, Pittsburgh, United States

**\*For correspondence:**
bwarecki@ucsc.edu (BW);
wtsulliv@ucsc.edu (WS)

†These authors contributed equally to this work

**Competing interest:** The authors declare that no competing interests exist.

**Abstract** *Wolbachia*, a vertically transmitted endosymbiont infecting many insects, spreads rapidly through uninfected populations by a mechanism known as cytoplasmic incompatibility (CI). In CI, a paternally delivered modification of the sperm leads to chromatin defects and lethality during and after the first mitosis of embryonic development in multiple species. However, whether CI-induced defects in later stage embryos are a consequence of the first division errors or caused by independent defects remains unresolved. To address this question, we focused on ~1/3 of embryos from CI crosses in *Drosophila simulans* that develop apparently normally through the first and subsequent pre-blastoderm divisions before exhibiting mitotic errors during the mid-blastula transition and gastrulation. We performed single embryo PCR and whole genome sequencing to find a large percentage of these developed CI-derived embryos bypass the first division defect. Using fluorescence in situ hybridization, we find increased chromosome segregation errors in gastrulating CI-derived embryos that had avoided the first division defect. Thus, *Wolbachia* action in the sperm induces developmentally deferred defects that are not a consequence of the first division errors. Like the immediate defect, the delayed defect is rescued through crosses to infected females. These studies inform current models on the molecular and cellular basis of CI.

## Editor's evaluation

This manuscript investigates the cellular and developmental defects underlying Wolbachia-induced cytoplasmic incompatibility (CI), which occurs when male insects harboring the endosymbiont bacteria Wolbachia fertilize eggs of uninfected females, triggering embryonic lethality. Previous work showed CI-induced defects early in embryogenesis (at first mitosis), whereas this work provides the detailed characterization of later defects including loss of nuclei ("nuclear fallout"), determination of haploidy versus diploidy in the bacteria-mediated lethality, and evidence that the mechanism of late embryonic defects is independent of the ones that drive early embryonic defects. The strength of evidence provided is compelling, including beautiful single embryo PCR analysis, and convincing light microscopy. This is a technically superb set of experiments. The significance of the findings is somewhat modest due to it being an incremental step forward, but it will be useful to those in the field.

## Introduction

*Wolbachia* are a bacterial endosymbiont present in the majority of insect species (*Weinert et al., 2015*; *Werren et al., 2008*). While they reside in the germline of both sexes, they are vertically transmitted exclusively through the female germline to all the offspring (*Kaur et al., 2021*). Consequently, *Wolbachia* have evolved a number of strategies that provide a selective advantage to infected females. This includes male killing, conversion of males to fertile females, induction of parthenogenesis, and most commonly cytoplasmic incompatibility (CI) (*Serbus et al., 2008*). CI is a form of *Wolbachia*-induced paternal effect embryonic lethality. Matings between infected males and uninfected females result in dramatic reductions in egg hatch rates (*Hoffmann et al., 1986*). However, matings between infected males and infected females, known as the 'rescue cross', results in normal egg hatch rates. Additionally, infected females mated with uninfected males results in normal hatch rates. Thus, in a *Wolbachia*-infected population, infected females have an enormous selective advantage over uninfected females as infected females produce normal hatch rates independent of the infection status of the male (*Turelli and Hoffmann, 1991*). This phenomenon, as well as *Wolbachia*-induced male sterility, is currently being employed throughout the world as a strategy for combating pest insects and insect-borne human diseases (*Jiggins, 2017*; *Moretti et al., 2018*; *Zheng et al., 2019*).

Since the discovery of CI and rescue (*Ghelelovitach, 1952*; *Yen and Barr, 1971*), there have been a number of insights into their molecular and cellular bases (*Shropshire et al., 2020*). Cytological studies demonstrate failures in condensation, alignment, and segregation of the paternal chromosomes during the first zygotic division in embryos derived from the CI cross (*Breeuwer and Werren, 1990*; *Callaini et al., 1997*; *Lassy and Karr, 1996*; *Reed and Werren, 1995*; *Ryan and Saul, 1968*; *Tram et al., 2006*; *Tram and Sullivan, 2002*). Subsequent studies demonstrated defects in the protamine-to-histone transition: deposition of the maternally supplied histone H3.3 is significantly delayed (*Landmann et al., 2009*). In addition, the male, but not the female, pronucleus of CI-derived embryos exhibits delays in DNA replication, nuclear envelope breakdown, and Cdk1 activation (*Tram and Sullivan, 2002*). As a result, passage of the male pronucleus through mitosis is delayed relative to the female pronucleus. Molecular insight into the mechanism of CI has come from recent studies demonstrating that a pair of *Wolbachia* genes originating from integrated viral DNA, the CI factors or Cifs, is likely responsible for CI (*Beckmann et al., 2017*; *LePage et al., 2017*). One of these genes, *cidB*, encodes a deubiquitylating enzyme and the other, *cinB*, a nuclease (*Chen et al., 2020*). When the gene pair is expressed in the male germline, paternal chromosome, and embryo abnormalities strikingly similar to *Wolbachia*-mediated CI are observed (*Beckmann et al., 2017*; *LePage et al., 2017*).

In addition to the well-characterized first division mitotic defects, studies in a number of species demonstrate that CI produces additional developmental defects and lethal phases later in embryogenesis (*Bonneau et al., 2018*; *Callaini et al., 1997*; *Callaini et al., 1996*; *Duron and Weill, 2006*; *Jost, 1970*; *Lassy and Karr, 1996*; *Wright and Barr, 1981*). Studies in wasps in which fertilized eggs develop into diploid females and unfertilized eggs develop into haploid males, provide insight into the different developmental outcomes in CI crosses (*Tram et al., 2006*). If CI disrupts but does not prevent paternal chromosome segregation, the resulting aneuploid embryos fail to develop. In contrast, if CI results in the complete failure of paternal chromosome segregation, embryos develop into haploid males bearing only the maternal chromosome complement (*Ryan and Saul, 1968*; *Tram et al., 2006*). In diplo-diploid organisms (where haploid embryos do not develop to adulthood), complete failure of paternal chromosome segregation in the first division leads to haploid embryos that develop and then subsequently fail to hatch (*Bonneau et al., 2018*; *Callaini et al., 1997*; *Callaini et al., 1996*; *Duron and Weill, 2006*; *Jost, 1970*). Additionally, in some organisms such as *Drosophila simulans*, a small fraction of CI-derived embryos do not undergo any first division defect and consequently hatch as diploids (*Lassy and Karr, 1996*).

In conjunction with broad developmental abnormalities, embryos developing from CI crosses also experience various cellular defects (*Callaini et al., 1996*). For example, studies in *Drosophila* have observed chromosome bridging in CI-derived embryos developing through the pre-blastoderm divisions (nuclear cycles 2–9) (*Lassy and Karr, 1996*; *LePage et al., 2017*). Additionally, irregular spindles are observed in syncytial and cellularized blastoderms (nuclear cycles 10–14) from CI crosses (*Callaini et al., 1996*). Other defects in developed embryos from CI crosses include displaced nuclei, clumped chromatin, and disorganized centrosomes in blastoderms and abnormally condensed nuclei

in gastrulating embryos (*Callaini et al., 1996*). However, due to inherent difficulties in visualizing fertilization and early embryonic development in *Drosophila*, whether the cellular defects observed in later stage embryos are the direct result of aneuploidy from first division errors or due to a poorly understood second, distinct set of CI-induced defects remained unresolved. Therefore, the extent and timing with which CI induces cellular defects throughout development is an open question.

Here, through a combination of live and fixed analysis, we directly address whether late developmental and cellular defects observed in CI-derived embryos are an outcome of the well-characterized first division errors or caused by a temporally distinct set of CI-induced cell cycle defects. Consistent with previous reports, we find that the majority of embryos derived from the CI cross arrest in the first division. However, we specifically focus our analyses on a population of about one-third of CI-derived embryos that apparently progress normally through the first zygotic and subsequent internal syncytial divisions. While these embryos undergo normal pre-blastoderm divisions, they exhibit significantly increased chromosome segregation defects during the mid-blastula transition, cellularization, and gastrulation. Sequence analysis reveals that ~40% of the CI-derived embryos that reach the blastoderm stage (>nuclear cycle 10) are diploid, having undergone a normal first division to develop with both maternal and paternal chromosomes. Using fluorescence in situ hybridization (FISH) to specifically select these diploid CI-derived embryos, we observe increased chromosome segregation defects during gastrulation, suggesting the late embryonic defects do not strictly require significant first division defects. Crosses to infected females (the rescue cross) reduce the frequencies of the induced division errors. These results reveal that *Wolbachia* CI-induced defects in the sperm produce developmentally deferred chromosome segregation defects in the late blastoderm divisions. Thus, our studies suggest CI produces temporally distinct early and late chromosome segregation defects. These findings provide insight into the mechanisms and timing of CI.

## Results

### *Wolbachia*-induced CI produces late embryonic lethality in ~30% of embryos

We used a combination of fixed and live analyses to determine the timing of defects as CI-derived embryos progressed through the early blastoderm divisions, cellularization, gastrulation, and hatching. We compared four different crosses: (1) the wild-type cross (uninfected male × uninfected female), (2) the CI-inducing cross (infected male × uninfected female), (3) the rescue cross (infected male × infected female), and (4) the reciprocal cross (uninfected male × infected female) (*Figure 1A*). Unless otherwise noted, we performed all experiments with *D. simulans* stocks that shared the same genetic background and were infected or uninfected with *Wolbachia* (*wRi*) (see Materials and methods). We used *D. simulans* because CI is particularly pronounced in this species and results in defects both during and after the first zygotic mitosis (*Callaini et al., 1997*; *Callaini et al., 1996*; *Lassy and Karr, 1996*).

To determine the timing of CI-induced embryonic lethal phases, we collected embryos from all four crosses and compared egg hatch rates to the percentage of embryos that had developed to at least nuclear cycle 10 (the syncytial blastoderm stage) (*Figure 1A, A'*). Consistent with previous results (*Hoffmann et al., 1986*), we observed a severe decrease in hatching for embryos derived from CI crosses (CI = 2%; N = 2397, compare to wild-type = 92%, N = 1208; rescue = 88%, N = 1281; reciprocal = 91%, N = 1299) (*Figure 1B, B'*). Thus, both CI-induced embryonic lethality and its corresponding rescue by maternally supplied *Wolbachia* are robust in *D. simulans*. Our analysis of fixed embryos revealed that the percentage of embryos that had developed to nuclear cycle 10 derived from wild-type (97%, N = 117), rescue (87%, N = 66), and reciprocal (100%, N = 47) crosses matched their respective hatch rates. However, unique to the CI cross, the percentage of CI-derived embryos that had developed to nuclear cycle 10 (28%, N = 159) was significantly higher than its hatch rate (2%, p = $2 \times 10^{-16}$ by $\chi$-squared test) (*Figure 1B, B'*). Therefore, a second lethal phase occurs at or after cortical nuclear cycle 10 that results in a significant proportion of the reduced egg hatch in CI-derived embryos.

To reduce any biological and environmental factors that could influence CI strength and embryonic development, we collected embryos from the same wild-type and CI crosses within 1 hr of each other (*Figure 1C*). We then analyzed the egg hatch rate with one set of embryos while fixing and DAPI

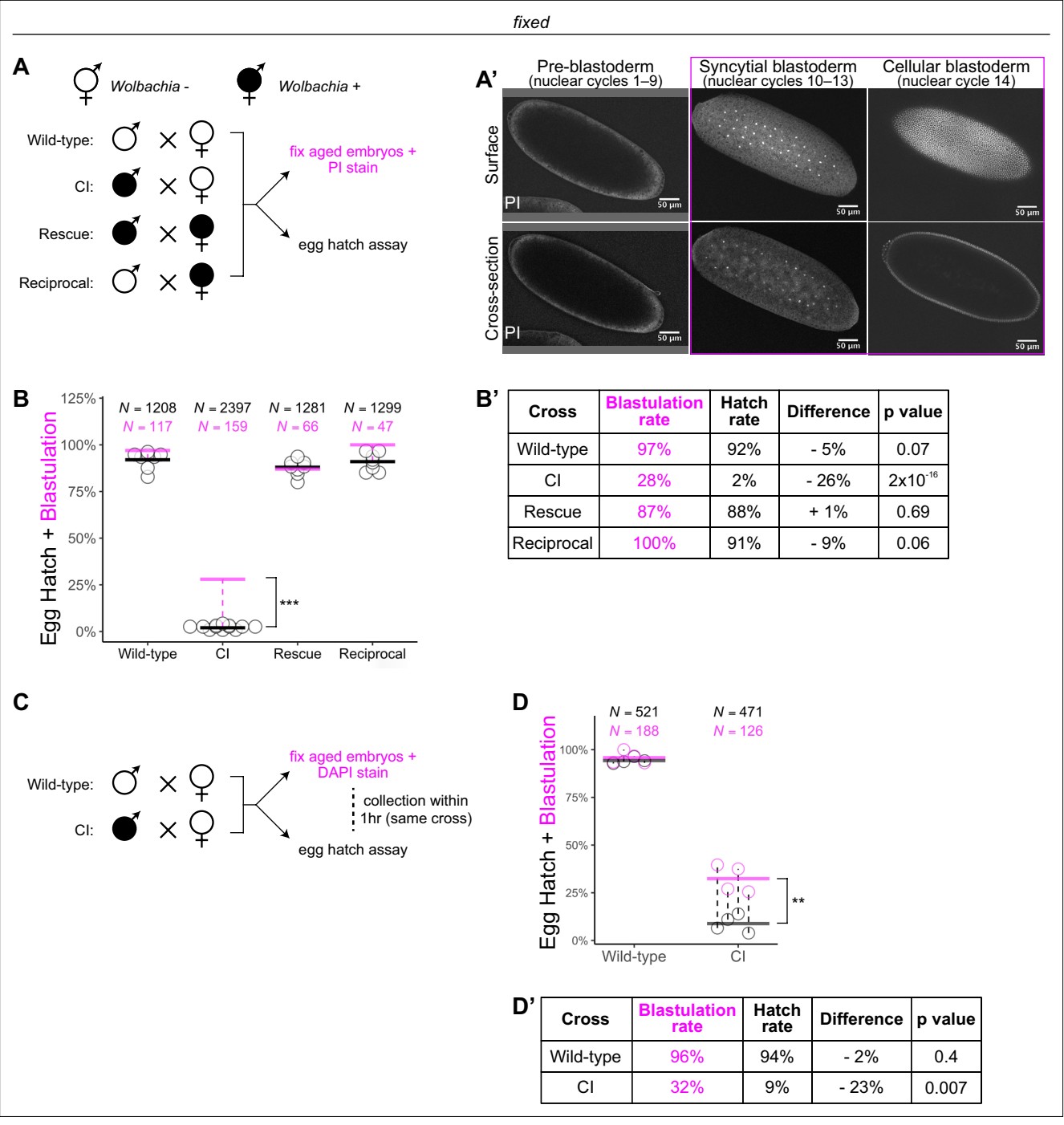

**Figure 1.** *Wolbachia* induces both early and late embryonic lethality. (**A**) *Wolbachia* infection status is indicated by filled circles. Embryos were collected from each of the four crosses and either used for egg hatch assays or aged prior to fixing and staining DNA with propidium iodide (PI). (**A'**) Confocal imaging of PI-stained embryos allowed categorization of embryo stage as pre-blastoderm (cycles 2–9), syncytial blastoderm (cycles 10–13), or cellular blastoderms (cycle 14). Scale bars are 50 μm. *** = p = 2 × 10⁻¹⁶ χ-squared test. (**B, B'**) Comparison between blastulation rate (% of fixed embryos staged as progressing beyond cycle 9) and egg hatch rate between each of the four crosses. Each circle represents one biological replicate of an egg hatch assay. Black and magenta lines represent the average egg hatch rate and the blastulation rate, respectively. *N* values correspond to the total number of embryos scored. While the hatch rate from wild-type, rescue, and reciprocal crosses closely corresponded to the blastulation rate, the hatch rate from cytoplasmic incompatibility (CI) embryos was statistically significantly decreased compared to the blastulation rate. Displayed p values were determined by χ-squared tests. (**C**) Embryos were collected from wild-type and CI crosses and were used to either determine embryo stage (by 4',6-diamidino-2-phenylindole (DAPI) staining) or egg hatch percentage in paired assays (collections were from the same crosses within 1 hr of each other). (**D, D'**) Comparison between blastulation rate (% of fixed embryos staged as progressing beyond cycle 9) and egg hatch for each cross. Each circle represents

*Figure 1 continued on next page*

*Figure 1 continued*

a technical replicate of one egg hatch assay and one staging experiment. Dashed lines connect paired experiments. Black and magenta lines represent the average egg hatch rate and the blastulation rate, respectively. *N* values correspond to the total number of embryos scored. The difference between blastulation rate and hatch rate was statistically significant by a two-sided paired *t*-test. ** = p = 0.007. (**D'**). See also *Figure 1—figure supplement 1*.

The online version of this article includes the following figure supplement(s) for figure 1:

**Figure supplement 1.** *Wolbachia* induces late embryonic lethality.

staining the other set to determine developmental stage. As before, the percentages of embryos that had developed to nuclear cycle 10 (96%, *N* = 188) and hatched (94%, *N* = 521) were similar in wild-type crosses (p = 0.4 by two-sided paired *t*-test) (*Figure 1D, D'*). In contrast, in CI crosses, the percentages of embryos that had developed to nuclear cycle 10 (32%, *N* = 126) were significantly higher than the percentages of embryos that hatched (9%, *N* = 471) (p = 0.007 by two-sided paired *t*-test) (*Figure 1D, D'*).

As an independent means of determining the lethal phases of embryos derived from the CI cross, we performed live analysis to compare the proportion of pre-blastoderm (nuclear cycles 2–9), syncytial blastoderm (nuclear cycles 10–13), or cellular blastoderm (nuclear cycle 14) in CI and wild-type crosses (*Figure 1—figure supplement 1*). Wild-type embryos developed to the syncytial blastoderm stage 91% (*N* = 40) of the time and hatched at a rate of 92% (*N* = 58). However, CI-derived embryos developed to the syncytial blastoderm stage 38% (*N* = 147) of the time but hatched at a significantly reduced rate of 16% (*N* = 110, p = $2 \times 10^{-4}$ by $\chi$-squared test) (*Figure 1—figure supplement 1*).

Therefore, consistent with previous results (*Bonneau et al., 2018*; *Callaini et al., 1997*; *Callaini et al., 1996*; *Duron and Weill, 2006*; *Lassy and Karr, 1996*), these data suggest at least two distinct lethal stages are associated with CI: the well-described lethal phase immediately following fertilization (~70% of embryos), and a second lethal phase that occurs well after the nuclei have undergone many rounds of syncytial and cellular mitoses (~30% of embryos). Significantly, rescue acts on both phases.

## Late-stage CI embryos initially develop normally through pre-blastoderm syncytial divisions before exhibiting mitotic defects and nuclear fallout during blastulation

In addition to late-stage lethality, CI-derived embryos exhibit cellular defects during later stages of development (*Callaini et al., 1997*; *Callaini et al., 1996*; *Lassy and Karr, 1996*; *LePage et al., 2017*). Since CI induces defective paternal chromosome segregation during the first embryonic division, which can result in either complete or partial loss of paternal chromosomes (*Bonneau et al., 2018*; *Landmann et al., 2009*; *Tram et al., 2006*), cellular defects in developed CI embryos may be due to holdover from errors during first division. One of the possible consequences of improper paternal chromosome segregation in the first division is daughter nuclei that inherit only part of the paternal chromosomes. This resulting segmental aneuploidy may then carry over into the subsequent mitoses (*Lassy and Karr, 1996*; *LePage et al., 2017*). Certainly, persistent DNA damage carried by the paternal chromatin could affect repeated syncytial divisions in the form of breakage–fusion–bridge cycles (*McClintock, 1941*; *Titen and Golic, 2008*).

Therefore, to assess any contribution of the first division segregation errors to late-stage CI-induced cellular defects, we examined fixed and DAPI-stained embryos in all stages of early embryonic development (nuclear cycles 2–14) (*Figure 2A*). For embryos fixed during nuclear cycles 2–9, we scored for anaphase bridging, unequally sized telophase daughter nuclei, and disorganized distributions of syncytial nuclei. For embryos fixed in cycles 10–14, we additionally scored for nuclear fallout, a process in which the products of defective divisions recede below the normal cortical monolayer of nuclei (*Sullivan et al., 1990*). As we were interested in the timing with which CI induces large-scale defects throughout development, we scored embryos as whole units, which potentially excludes minor defects.

As expected, wild-type-derived embryos exhibited abnormalities in 0% (0/64) of syncytial pre-blastoderm divisions (cycles 2–9), 0% (0/13) of early cortical divisions (cycles 10–11), and 2% (1/58) of late cortical divisions (cycles 12–14) (*Figure 2B*). Similarly, CI-derived embryos exhibited abnormalities in only 3% (2/63) of syncytial pre-blastoderm divisions (cycles 2–9). However, we observed a significant increase in CI-derived embryos with abnormal nuclei during early cortical divisions (cycles

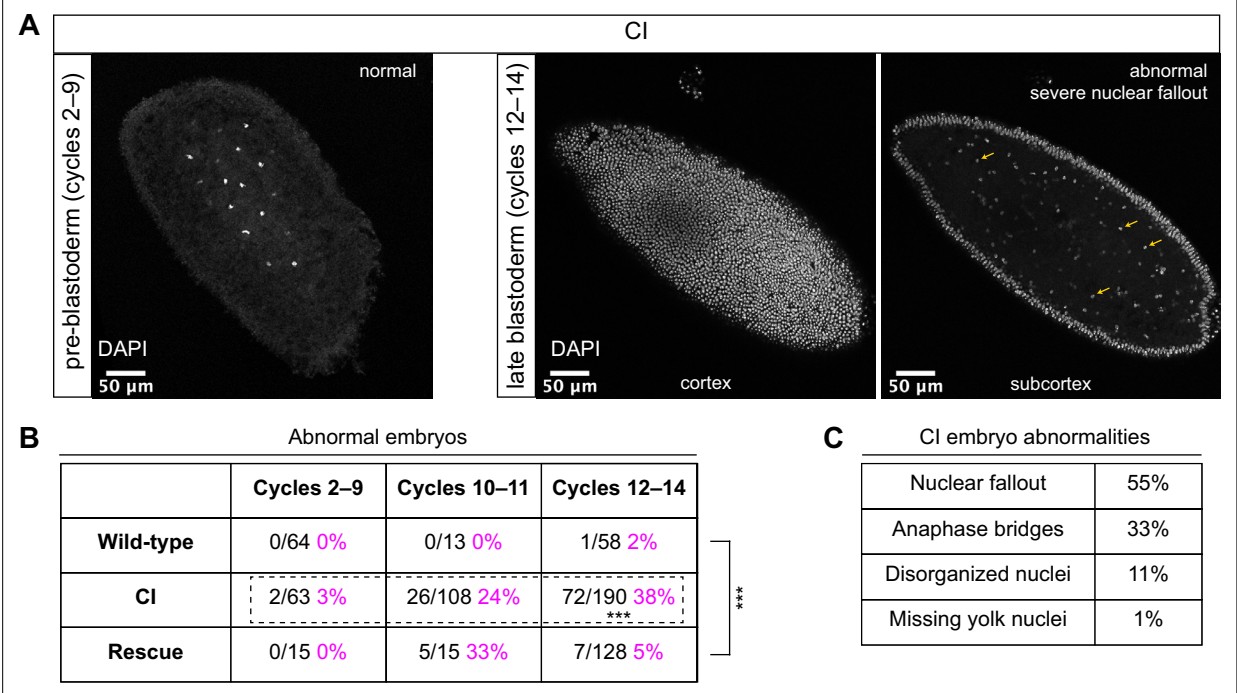

**Figure 2.** Cytoplasmic incompatibility (CI)-derived embryos proceed normally through pre-blastoderm divisions and then exhibit cellular defects during blastoderm divisions. (**A**) Examples of fixed and DAPI-stained CI-derived embryos from pre-blastoderm (cycles 2–9) and late blastoderm (12–14) stages. While the CI-derived pre-blastoderm appears normal, the late blastoderm exhibits severe nuclear fallout (nuclei receded from the cortex and into subcortical regions). Arrows point to several examples of fallen out nuclei. Scale bars are 50 µm. (**B**) Comparison of the percentage of abnormal embryos from wild-type, CI, and rescue crosses during different stages of embryogenesis. While CI-derived embryos developed normally through cycles 2–9, they exhibited significantly increased abnormalities during cycles 10–14 and 12-14 (\*\*\* = p = 1.9 x 10⁻⁸ by two-sided Fisher's exact test). Abnormalities in cycles 12–14 were significantly reduced in embryos from rescue crosses (\*\*\* = p = 3.8 x 10⁻¹⁶ by two-sided Fisher's exact test). *N* values correspond to the number of embryos scored (biological replicates). (**C**) Classification of abnormalities observed in CI-derived embryos.

10–11, 24%, 26/108) and late cortical divisions (cycles 12–14, 38%, 72/190) (p = $1.9 \times 10^{-8}$ by two-sided Fisher's exact test) (*Figure 2B*). We regularly observed nuclear fallout (55%; *Figure 2*, yellow arrows), anaphase bridging (33%), and disorganized nuclei (11%) in CI-derived cycle 10–14 embryos (*Figure 2C*). Additionally, we found that the CI-induced increase in abnormal nuclear divisions during late cortical divisions (cycles 12–14) was dramatically, but not completely, reduced in the rescue cross (5%, 7/128) (p = $3.8 \times 10^{-16}$ by two-sided Fisher's exact test) (*Figure 2B*).

Thus, CI-derived embryos that bypass the first lethal phase develop normally through nuclear cycles 2–9 and then exhibit a dramatic increase in abnormal segregation and nuclear organization during the cortical nuclear cycles (10–14). The normal development through nuclear cycles 2–9 suggests that the cortical division defects observed in the CI-derived embryos are not a direct consequence of abnormal first divisions but may instead be separate CI-induced defects. Significantly, as with the first division CI-induced defects, CI-induced cortical defects are rescued when infected males are crossed to infected females.

## Blastoderm embryos from CI crosses have higher rates of nuclear fallout than embryos from wild-type or rescue crosses

To further explore the defects that arise in blastoderm CI embryos, we performed a more sensitive assay to score the number of abnormal cortical nuclei that recede into the interior of the embryo, known as nuclear fallout. Because the fidelity of cortical nuclear divisions is maintained by a mechanism that eliminates the products of abnormal divisions from the cortex (*Sullivan et al., 1990*), assaying nuclear fallout provides a quantitative measure of cortical division errors (*Sullivan et al., 1993*). This assay is sensitive due to the lack of gap phases in cortical divisions. Even in wild-type embryos, nuclear fallout is observed at a low level (*Sullivan et al., 1993*). Consequently, we used this assay to determine the effect of *Wolbachia*-induced CI on the cortical syncytial divisions.

We regularly observed CI embryos with increased numbers of nuclei that had fallen from the cortical layer of nuclei into the subcortex (*Figure 3*, magenta arrows). We quantified the amount of nuclear fallout per cycle 10–14 embryo from wild-type (1.3 ± 2.4, $N$ = 85), CI (6.6 ± 6.4, $N$ = 34), rescue (1.7 ± 3.8, $N$ = 60), and reciprocal (0.9 ± 1.4, $N$ = 35) crosses (*Figure 3B, B′*). The amount of nuclear fallout per embryo was significantly increased in CI embryos compared to wild-type embryos (p = $4.5 \times 10^{-8}$ by Mann–Whitney test) and to rescue embryos (p = $1.7 \times 10^{-6}$ by Mann–Whitney test). Given the increased nuclear density during the final blastoderm cycle, we observed a more pronounced increase in nuclear fallout in cycle 14 CI embryos (CI = 11.7, $N$ = 14; wild-type = 1.0, $N$ = 64) (*Figure 3B′*).

The above analyses excluded the extreme posterior region of the embryo. This is because in wild-type embryos, the extreme posterior region is composed of 8–10 cellularized pole cells that have migrated to the cortex ahead of the main contingent of dividing nuclei. These cells are the precursors to the germline (*Illmensee and Mahowald, 1974*). In general, cortical nuclei in this posterior region exhibit a higher rate of nuclear fallout compared to somatic nuclei in the rest of the embryo (*Figure 3A*, yellow arrows). Similar to nuclear fallout in the rest of the embryo, nuclear fallout in the posterior pole was dramatically higher in cycle 10–14 embryos from CI crosses compared to those from wild-type (p = $6.8 \times 10^{-4}$ by Mann–Whitney test), rescue (p = $2.5 \times 10^{-3}$ by Mann–Whitney test), and reciprocal crosses (*Figure 3C, C′*).

Previous work has shown that nuclear fallout occurs via detachment of the cortical nuclei from their centrosomes (*Sullivan et al., 1993*). To determine if the nuclear fallout in embryos from CI crosses is due to a similar detachment from centrosomes, we next fixed embryos from CI crosses and costained with DAPI and antibodies that recognize centrosomin, a key component of centrosomes (*Megraw et al., 1999*; *Figure 3D*). Receding nuclei create a gap in the normally dividing cortical surface nuclei. The centrosomes associated with the fallen nuclei (green arrows) remained on the cortex (*Figure 3D*). Thus, nuclei in CI embryos regularly detach from their centrosomes and recede from the cortex.

## Lagging chromosomes are a proximal cause of nuclear fallout in CI-derived embryos

To determine the primary cause of the errors leading to nuclei falling out from the cortical monolayer of normally dividing nuclei, we injected early embryos with rhodamine-labeled histones and performed live imaging on a confocal microscope. Live analysis allowed us to identify receding nuclei and analyze the proximal mitotic defects that led to nuclear fallout. For both CI- and wild-type-derived embryos, we observed nuclear fallout occurred almost exclusively during the telophase-to-interphase transition (*Figure 4*). This is likely the result of a failure of the nuclei to maintain association with their centrosomes following errors in the preceding division.

We routinely observed that nuclear fallout in telophase/interphase was immediately preceded by defective sister chromatid separation during anaphase (*Figure 4B*, *Figure 4—video 1*). While nuclei in which sister chromatids had segregated normally (*Figure 4B*, magenta circles) remained on the surface and entered the next cell cycle, nuclei in which sister chromatids had severely lagged (*Figure 4B*, yellow and blue circles) immediately receded into the interior of the embryo during the subsequent interphase. In total, 70% of nuclei destined to fallout in CI-derived embryos were preceded by lagging or bridged chromosomes, as in wild-type-derived embryos (*Figure 4C*).

Defects causing segregation errors in nuclei destined to fallout may also result in the activation of the spindle-assembly checkpoint that would have subsequently delayed entry into anaphase. Therefore, we compared the timing of the metaphase-to-anaphase transition in divisions that resulted in fallout to those of their neighboring normal divisions (*Figure 4D*). Only a small fraction of nuclei destined to fallout (22%) exhibited a delay in anaphase entry compared to their neighboring nuclei ('late'). In contrast, the vast majority entered anaphase synchronously with (74%, 'on-time') or preceding (4%, 'early') their neighbors. Interestingly, in wild-type embryos, a greater fraction (40%) of nuclei destined to fallout delayed metaphase exit. Therefore, we were unable to regularly detect spindle-assembly checkpoint-mediated delays in CI embryos at our level of temporal resolution.

## Defective chromosome segregation persists after cellularization in CI-derived embryos

Following completion of nuclear cycle 13, in an event known as cellularization, each syncytial nucleus is encompassed by an ingressing plasma membrane resulting in the simultaneous formation of individual

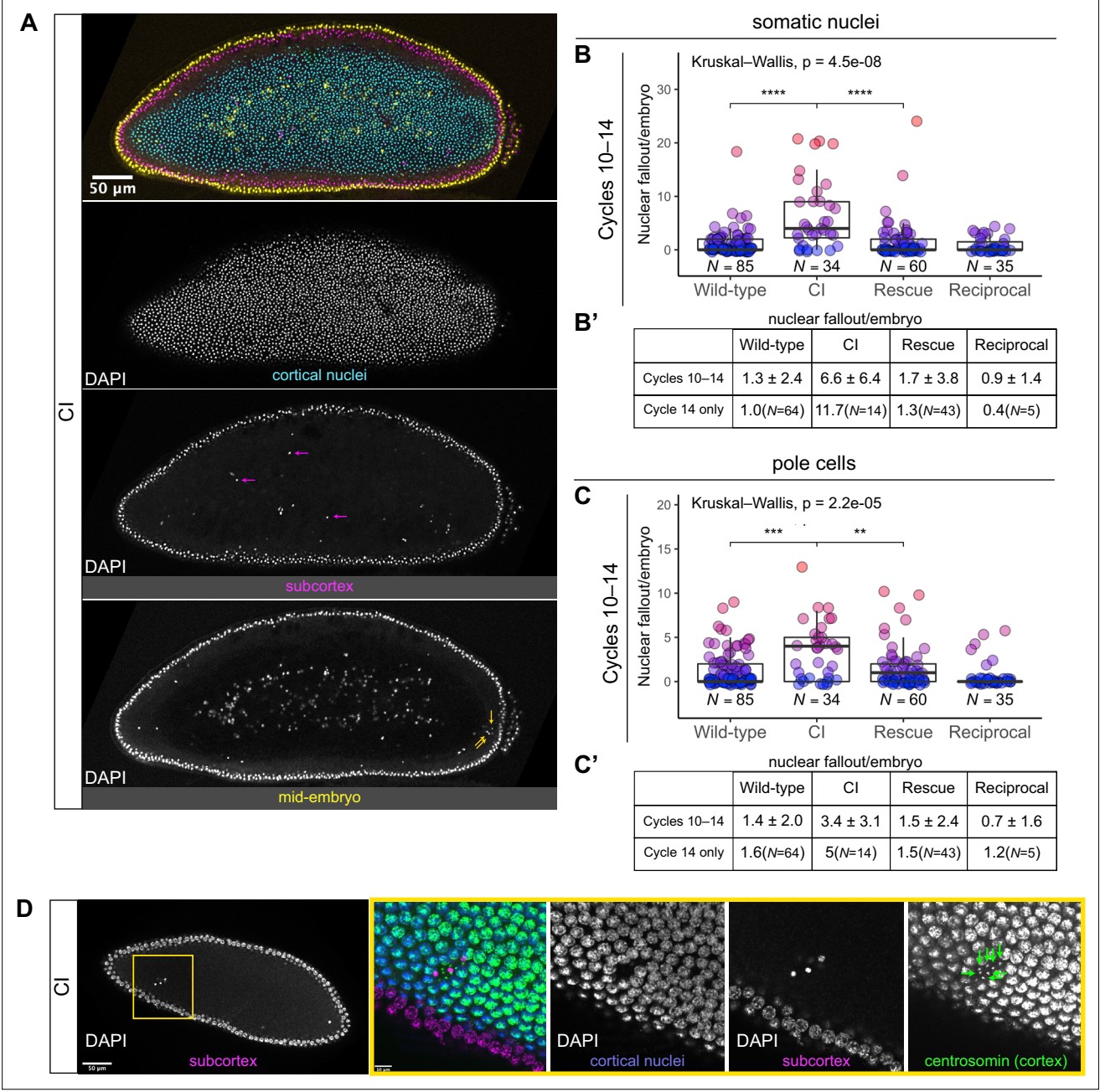

**Figure 3.** Developed cytoplasmic incompatibility (CI)-derived embryos exhibit increased rates of nuclear fallout. (**A**) Image of a CI-derived blastoderm exhibiting moderate nuclear fallout. Cortical nuclei (cyan) are on the surface of the embryo. Nuclei that have fallen out of the cortex can clearly be observed 5–10 µm beneath the cortex (subcortex, magenta) and at the mid-plane of the embryo (yellow). Magenta arrows point to examples of somatic nuclei that have fallen out. Yellow arrows point to examples of pole cells that have fallen out. Scale bar is 50 µm (**B, B'**) Comparison of somatic nuclear fallout in cycle 10–14 embryos from wild-type, CI, rescue, and reciprocal crosses. (**B**) Each dot represents the number of fallen nuclei per embryo. Asterisks indicate significance by Mann–Whitney tests (Wild-type to CI: p = 4.5 x 10⁻⁸; CI to rescue: p = 1.7 x 10⁻⁶). (**B'**) Averages and standard deviations are summarized. CI-derived embryos have significantly increased somatic nuclear fallout compared to wild-type- and rescue-derived embryos. N values correspond to the number of embryos scored (biological replicates). (**C, C'**) Comparison of pole cell nuclear fallout in cycle 10–14 embryos from wild-type, CI, rescue, and reciprocal crosses. (**C**) Each dot represents the number of fallen nuclei per embryo. Asterisks indicate significance by Mann–Whitney tests (Wild-type to CI: p = 6.8 x 10⁻⁴; CI to Rescue: p = 2.5 x 10⁻³). (**C'**) Averages and standard deviations are summarized. CI-derived embryos have significantly increased somatic nuclear fallout compared to wild-type- and rescue-derived embryos. N values correspond to the number of embryos scored (biological replicates). (**D**) CI-derived embryo stained with anti-centrosomin antibody to mark centrosomes and counterstained with DAPI. While cortical nuclei (blue) remain strongly associated with their centrosomes (green), nuclei that recede into the subcortex (magenta) detach from their centrosomes that are left at the cortex (green arrows). Yellow box indicates zoomed region. Scales bars are 50 and 10 µm for unzoomed and zoomed regions, respectively.

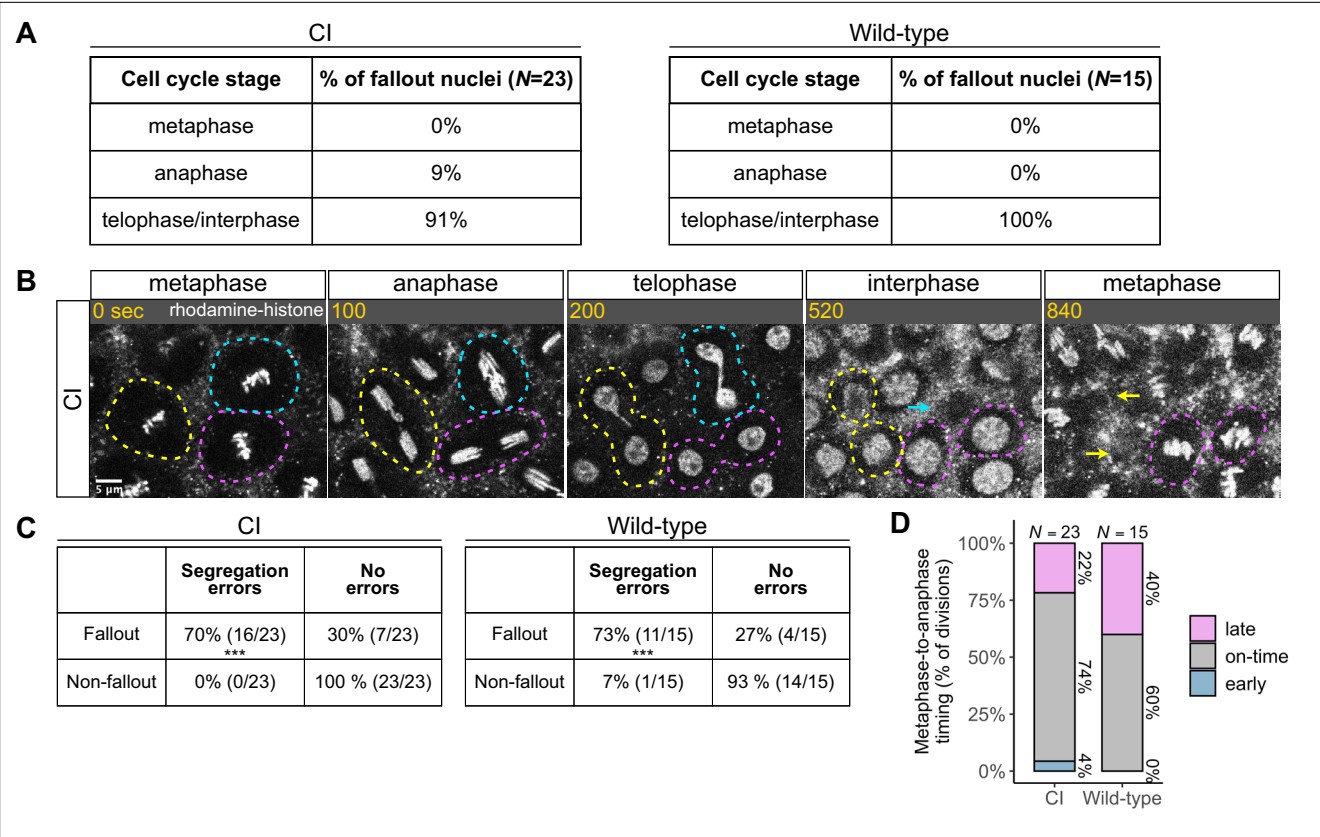

**Figure 4.** Chromosome segregation errors are the proximate cause of nuclear fallout in cytoplasmic incompatibility (CI)-derived embryos. (**A**) Comparison of when in the cell cycle nuclei fallout in both CI- and wild-type-derived embryos. (**B**) Nuclei that fallout (yellow and blue circles) exhibit severely lagging chromosomes in the previous division, while nuclei that remain at the cortex (magenta circle) exhibit normal chromosome segregation. Scale bar is 5 μm, and time is written in seconds. See also *Figure 4—video 1*. (**C**) Chromosome segregation errors were significantly more frequent for nuclei destined to fallout compared to neighboring nuclei (non-fallout) in both CI- (*** = p = 3.4 × 10⁻⁶ by χ-squared test) and wild-type-derived embryos (*** = p = 8.0 × 10⁻⁴ by χ-squared test). (**D**) Comparison of metaphase-to-anaphase timing between nuclei destined to fallout and their neighbors that remain at the cortex in both CI- and wild-type-derived embryos. 'Early' = fallout nuclei enter anaphase before their neighbors. 'On-time' = fallout and neighboring nuclei enter anaphase simultaneously. 'Late' = fallout nuclei enter anaphase after their neighbors. *N* values correspond to the number of cells scored (biological replicates). See also *Figure 4—figure supplement 1*.

The online version of this article includes the following video and figure supplement(s) for figure 4:

**Figure 4—video 1.** Chromosome segregation immediately precedes nuclear fallout of cortical blastoderm nuclei.
https://elifesciences.org/articles/81292/figures#fig4video1

**Figure supplement 1.** Cytoplasmic incompatibility (CI)-derived embryos exhibit increased rates of chromosome segregation errors and nuclear fallout.

cells (*Sokac et al., 2022*). After cellularization, gastrulation begins (*Foe, 1989*). Invaginations form the head furrow, and bilateral groups of cells throughout the embryo, referred to as mitotic patches, undergo another round of mitosis. We reasoned that CI-induced segregation defects might persist in these mitoses following cellularization.

To examine if chromosome segregation defects occur in CI-derived embryos after the establishment of individual cells, we fixed and DAPI-stained gastrulating embryos from wild-type, CI, and rescue crosses (*Figure 4—figure supplement 1A*) and quantified the frequency of division errors in each cross (*Figure 4—figure supplement 1B–D'*). While chromosome segregation defects in gastrulating embryos from wild-type crosses occurred at a low rate (11%, *N* = 321 divisions/25 embryos), CI-derived embryos exhibited a significant increase in segregation defects (34%, *N* = 687 divisions/40 embryos) (p = 7.7 × 10⁻⁷ by Mann–Whitney test) (*Figure 4—figure supplement 1D, D'*). Significantly, we observed a reduction in segregation errors in embryos from the rescue cross (19%, *N* = 485 divisions/30 embryos) (p = 8.6 × 10⁻⁴ by Mann–Whitney test), although the level of segregation errors was still increased compared to that of wild-type embryos (p = 0.009 by Mann–Whitney test).

Thus, CI-derived embryos exhibit increased chromosome segregation errors that begin in blastoderm stages and continue post-cellularization.

## ~60% of CI-derived blastoderm embryos are haploid and strongly associated with embryonic lethality, while the remaining ~40% of are diploid

Previous studies have linked late embryonic lethality to haploid development arising from CI-induced chromosome segregation defects during the first division (*Bonneau et al., 2018*; *Callaini et al., 1997*; *Duron and Weill, 2006*). Should CI be strong, paternal chromosomes are completely eliminated during the first division, and embryos develop bearing only the maternal chromosome complement (*Tram et al., 2006*). In diplo-diploid organisms, these haploid embryos then die before hatching (*Bonneau et al., 2018*; *Callaini et al., 1997*; *Duron and Weill, 2006*). Our observation of mitotic defects in CI-derived blastoderm and gastrulating embryos offers a potential additional explanation for late embryonic lethality. Therefore, we wished to reexamine the relationship between complete paternal chromosome exclusion resulting in haploidy and late embryonic lethality.

To assess the relationship between haploidy and lethality in CI-derived embryos, we performed CI and rescue crosses in which the *Wolbachia*-infected fathers transmitted an *egfp* transgene to all their offspring (*Figure 5A*). The resulting embryos from these crosses should be genotypically identical. We additionally performed wild-type crosses with uninfected fathers bearing no transgene, serving as a negative control (*Figure 5A*). We selected embryos that developed to the cellular blastoderm stage and performed single embryo PCR with primers complementary to the paternally transmitted *egfp* transgene. The *egfp* transgene was always detected in embryos from the positive control rescue cross (~1.4 kb band) and never detected in our negative control embryos derived from uninfected males lacking the *egfp* transgene (*Figure 5B*). In contrast, we only detected the *egfp* transgene in 42% (*N* = 91) of CI-derived cellular blastoderms (*Figure 5B, C*). This indicates that while many of the CI-derived blastoderm embryos are diploid, a significant proportion of late-stage CI embryos are haploid. We regularly observed that the overall percentage of *egfp*-positive, diploid embryos correlated with the percentage of hatched eggs from paired egg hatch assays (*Figure 5C'*), suggesting only the *egfp*-negative, haploid embryos were failing to hatch. Thus, as previously reported (*Bonneau et al., 2018*; *Callaini et al., 1997*; *Duron and Weill, 2006*), haploidy due to loss of paternal chromosomes is linked with late embryonic lethality.

## Late-stage defects are not due to large-scale chromosome fragmentation and mosaicism after the first division

Although we found haploidy to be strongly associated with late embryonic lethality, haploidy does not inherently cause the type of chromosome segregation errors we regularly observed in late CI-derived embryos (*Debec, 1978*; *Tang et al., 2017*). An alternative potential cause for the segregation errors characterized here is segmental aneuploidy due to an incomplete exclusion of the paternal chromosomes during the first division that does not disrupt early embryonic development. In this scenario, partial chromosome loss or chromosome fragmentation is transmitted from the first division through seemingly normal syncytial divisions and then causes the segregation errors seen in later developmental stages.

To test the possibility that fragmented paternal chromosomes are transmitted through the syncytial divisions, we sequenced the entire genome of individual cellular blastoderms and then mapped read depth to five specifically chosen coding regions at different locations for each major chromosome or chromosome arm (*Figure 5D, E*). As above, males in the CI and rescue crosses were homozygous for the *egfp* transgene, allowing us to distinguish between embryos in which paternally derived chromosomes were either present or absent. For each individual embryo, we then compared read depth at each locus to the mean read depth for the embryo's entire genome. Comparing multiple coding regions on each chromosome arm allowed us to detect large-scale changes in chromosome copy number despite inherent variation in the read depth for single genes in individual embryos. For example, we regularly detected ½ the mean genome depth for reads mapping to coding regions on the X and Y chromosomes in XY male embryos (*Figure 5E*). While we cannot rule out small deletions, these data suggest that neither haploid (*egfp*-negative) nor diploid (*egfp*-positive) CI-derived embryos exhibited large-scale chromosome loss consistent with the mitotic transmission of only part

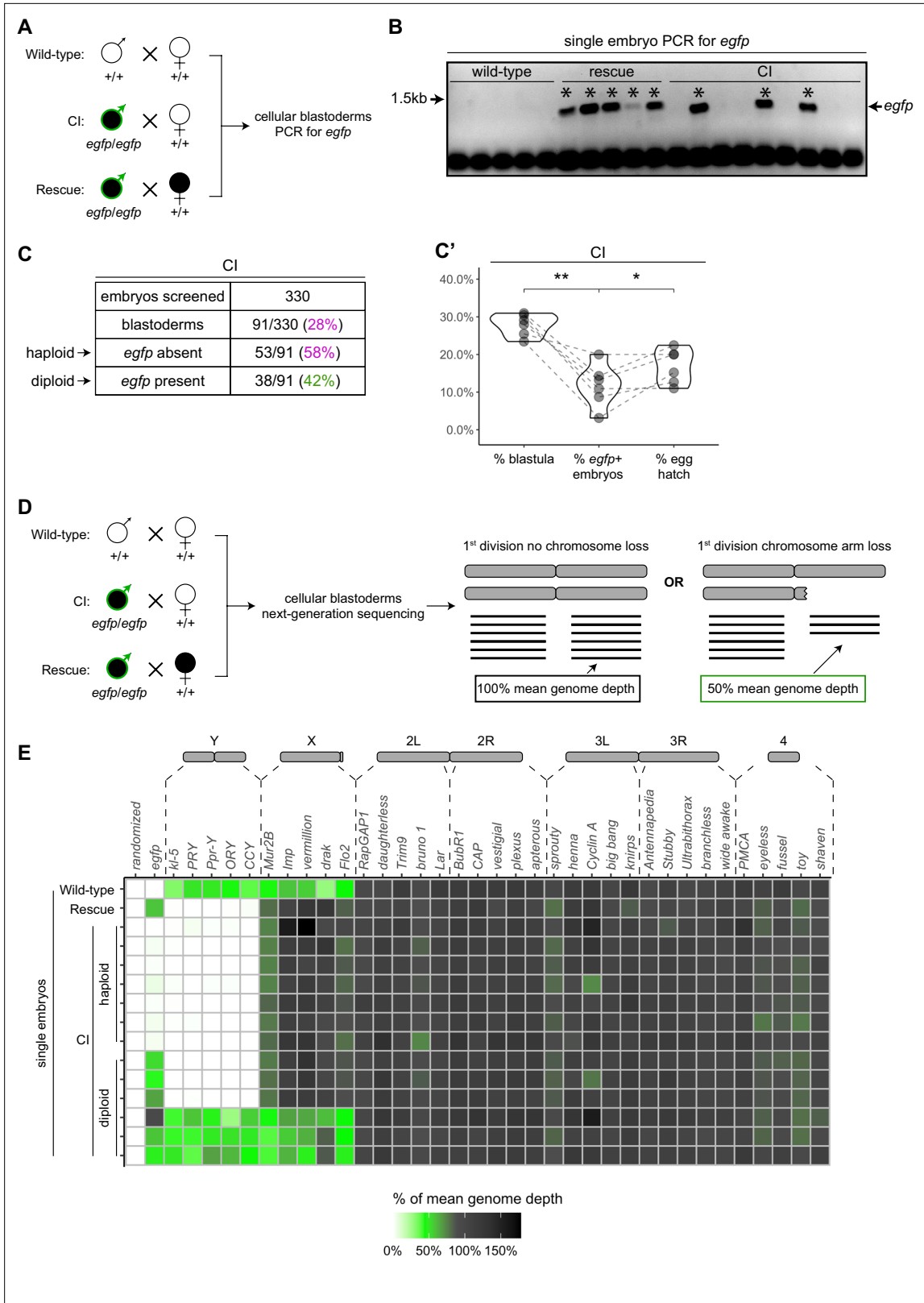

**Figure 5.** Late-stage cytoplasmic incompatibility (CI)-derived embryos are either haploid (maternal chromosome set) or diploid (both parental chromosome sets). (**A**) Embryos were collected from wild-type crosses or CI and rescue crosses in which the father was homozygous for an *egfp* transgene. Embryos were staged live, and cellular blastoderms were selected for single embryo PCR analysis with primers recognizing *egfp*. (**B**) A representative gel showing detection of *egfp* (asterisks) in all rescue-derived embryos and in a mix of CI-derived embryos. No *egfp* is detected in the

*Figure 5 continued on next page*

*Figure 5 continued*

wild-type control. See also *Figure 5—source data 1*. (**C**) Summary of the percentage of screened CI-derived cellular blastoderms in which either *egfp* was absent (haploid) or *egfp* was present (diploid). *N* values correspond to the number of embryos scored (biological replicates). (**C'**) Comparison of the percentages of embryos that had reached at least cycle 10 (% blastula), had detectable *egfp* bands (% *egfp* + embryos) and a concomitant egg hatch (% egg hatch). Each dot represents one technical replicate, and lines connect values for the same experiment. The percentage of *egfp* + embryos (diploids) was more associated with the percentage of eggs that hatched (p = 0.045 by Mann–Whitney test) than with the percentage of blastoderms screened (p = 0.001 by Mann–Whitney test), suggesting haploid embryos do not hatch. (**D**) Embryos were collected from wild-type crosses or CI and rescue crosses in which the father was homozygous for an *egfp* transgene. Embryos were staged live, and cellular blastoderms were selected for single embryo sequencing. If chromosome arms were not lost during the first division, the mean depth of reads mapping to the chromosome arm should be near the mean depth of reads mapping to the genome (black box). If chromosome arms were lost during the first division, the mean depth of reads aligning to that chromosome arm should be 50% of the mean depth of reads aligning to the genome (green box). In haploids, maternal chromosome arm loss would result in no reads mapping to that chromosome arm. (**E**) Sequenced embryos were sorted as haploid or diploid based on the depth of reads mapping to *egfp*. Each box represents the mean depth of reads aligning to that gene divided by the mean depth of reads aligning to the whole genome ('mean genome depth'). White = 0% of mean genome depth; green = 50% of mean genome depth; gray = 100% of mean genome depth; black = 150% of mean genome depth. Consistent with no partial chromosome/chromosome arm loss, genes across all chromosomes were present at 100% mean genome depth for both haploids and diploids (or 50% for X-linked genes when embryos are male). See also *Figure 5—figure supplement 1*. See also *Figure 5—source data 2*.

The online version of this article includes the following source data and figure supplement(s) for figure 5:

**Source data 1.** Raw gel images for single embryo PCR assay *egfp* presence/absence.

**Source data 2.** Normalized depth of reads aligning to coding sequences and *egfp* in wild-type-, cytoplasmic incompatibility (CI)-, and rescue-derived cellular blastoderms.

**Figure supplement 1.** Depth of reads aligning to genome in cytoplasmic incompatibility (CI)-derived cellularized embryos is decreased compared to wild-type- and rescue-derived embryos.

of the paternal genome (*Figure 5E*, *Figure 5—source data 2*). Instead, both haploid and diploid embryos had full euploid sets of chromosomes corresponding to either 1n or 2n, respectively. This suggests that late-stage CI embryos had either (1) lost all their paternal chromosomes during the first division or (2) did not experience any significant chromosome loss during the first division at all. Thus, defects observed in late-stage CI embryos cannot be due to partial chromosome loss or fragmentation carried over from the first division.

A separate, potentially interesting, outcome of this experiment is that we found CI-derived embryos regularly had less depth of reads mapping to their entire genome than either wild-type or rescue embryos (*Figure 5—figure supplement 1*). This was true for both haploid and diploid CI embryos. Normalizing the depth of reads aligning to the whole genome to the depth of reads aligning to the mitochondrial genome (which should be unchanged) for each embryo suggested differences in sequencing input may not fully explain the decrease in reads mapping to CI embryos (*Figure 5—figure supplement 1C*). Although we cannot exclude how any variation in sequencing multiple samples may affect these results, this finding raises the intriguing possibility of intrinsic differences in the chromatin of CI and wild-type-derived blastoderm embryos.

## Late-stage mitotic errors in diploid CI-derived embryos are due to a second CI-induced defect temporally distinct from the first division defect

Given neither haploidy nor chromosome fragmentation arising from the first division defect explains the mitotic errors we observed in CI-derived blastoderms and gastrulating embryos, we hypothesized that there is instead a second, CI-induced defect completely separate from the first division defect. To test this hypothesis, we asked if CI embryos that had completely 'escaped' the first division defect had increased mitotic errors during later developmental stages.

As described above, late-stage embryos are either haploid, missing their complete paternal chromosome complement, or diploid, having escaped any first division defect to develop with both maternal and paternal chromosome sets (*Figure 5E*). These diploid embryos can be identified by the presence of a paternally derived chromosome, such as the Y chromosome, which is only detectable in diploids and never in haploids (*Figure 5E*). The *D. simulans* Y chromosome can be identified by FISH with Y-specific probes (*Ferree and Barbash, 2009*). Therefore, we fixed gastrulating embryos from wild-type, CI, and rescue crosses, labeled the Y chromosome with fluorescent probes to select

embryos that had escaped the first division defect, counterstained with DAPI to score any mitotic defects.

While Y-bearing gastrulating embryos from wild-type crosses (*Figure 6*) exhibited relatively normal chromosome segregation, we observed lagging and bridging chromosomes in Y-bearing embryos from CI crosses (*Figure 6B*, white arrow). Additionally, in Y-bearing embryos from rescue crosses, chromosome segregation proceeded normally (*Figure 6C*). The increase in chromosome segregation errors in Y-bearing CI-derived embryos (15%, $N$ = 1095 divisions/23 embryos) compared to Y-bearing wild-type embryos (7%, $N$ = 1758 divisions/45 embryos) was statistically significant (p = $6.7 \times 10^{-6}$ by Mann–Whitney test) (*Figure 6D, D'*). As the diploid, Y-bearing embryos had completely escaped the first division defects, these results demonstrate that late-stage mitotic errors are due to a second CI-induced defect independent of the first division defect. The reduction of chromosome segregation errors in Y-bearing rescue-derived embryos (9%, $N$ = 985 divisions/32 embryos) compared to Y-bearing CI-derived embryos was also statistically significant (p = $1.1 \times 10^{-3}$ by Mann–Whitney test) (*Figure 6D, D'*), indicating maternally supplied *Wolbachia* also rescue this second defect.

Although these CI-derived embryos are diploid and are likely to hatch despite the observed division defects, we found a subsequent decrease in the rate of hatched eggs that develop into adults in CI crosses compared to both wild-type and rescue crosses (*Figure 6—figure supplement 1*). We collected embryos from wild-type, CI, and rescue crosses and first performed egg hatch assays. Next, we determined the number of hatched eggs from these assays that then developed into adults (*Figure 6—figure supplement 1*). Similar to the increased lethality in embryos derived from CI crosses, we also observed statistically significant increased lethality in larvae derived from CI crosses compared to larvae derived from wild-type crosses (*Figure 6—figure supplement 1*, 'hatched egg-to-adult', p = 0.008 by Mann–Whitney test). This larval lethality was significantly reduced in the rescue cross (*Figure 6—figure supplement 1*, p = 0.03 by Mann–Whitney test). In total, out of 966 eggs collected from CI crosses, 137 hatched. Of those 137 hatched eggs, only 94 (69%) developed into adults (*Figure 6—figure supplement 1*). In contrast, 520 of 548 (95%) and 511/588 (87%) of hatched eggs from wild-type and rescue crosses, respectively, developed into adults (*Figure 6—figure supplement 1*). For the progeny of CI crosses, relatively stronger embryonic lethality was often associated with relatively stronger larval lethality (*Figure 6—figure supplement 1*). Therefore, *Wolbachia* action in the sperm may induce remarkably deferred CI defects that contribute to the selective advantage of infected females by promoting increased lethality during post-hatching development.

## Discussion

In addition to the well-characterized early embryonic arrest, a number of reports reveal a large portion of CI-derived embryos undergo substantial embryonic development but then fail to hatch (*Bonneau et al., 2018*; *Callaini et al., 1997*; *Callaini et al., 1996*; *Duron and Weill, 2006*). In addition, these late-developing CI-derived embryos also exhibit significant cellular defects including chromosome segregation errors, irregular spindles, displaced nuclei, and disorganized centrosomes (*Callaini et al., 1997*; *Callaini et al., 1996*; *Lassy and Karr, 1996*). In diplo-diploid species, late embryonic lethality is widely believed to be due to haploid development explained by the behavior of the paternal chromosomes during the first division (*Bonneau et al., 2018*; *Callaini et al., 1997*; *Duron and Weill, 2006*). While weak CI results in defective paternal chromosome segregation creating aneuploid nuclei that arrest in early embryonic development, strong CI results in complete failure of sister chromosome segregation and haploid nuclei bearing only the maternal chromosome complement (*Bonneau et al., 2018*; *Callaini et al., 1997*; *Duron and Weill, 2006*; *Tram et al., 2006*). Haploid embryos develop normally to the cellular blastoderm stage but then fail prior to hatching.

However, whether the numerous cellular defects observed in late-developing CI-derived embryos are also the result of first division errors has not been resolved. Here, we addressed this question by analyzing the timing, extent, and causes of mitotic defects observed specifically in late *D. simulans* embryos derived from CI crosses that seemingly bypass the first division defects and develop normally through early embryogenesis. Collectively, our results suggest that late embryonic defects are the result of a second CI-induced affect temporally distinct from first division errors. As discussed below, this suggests CI may be produced through either (1) a common acute mechanism that acts at both early and late development time points, or (2) separate and distinct early and late mechanisms that do not depend upon one another.

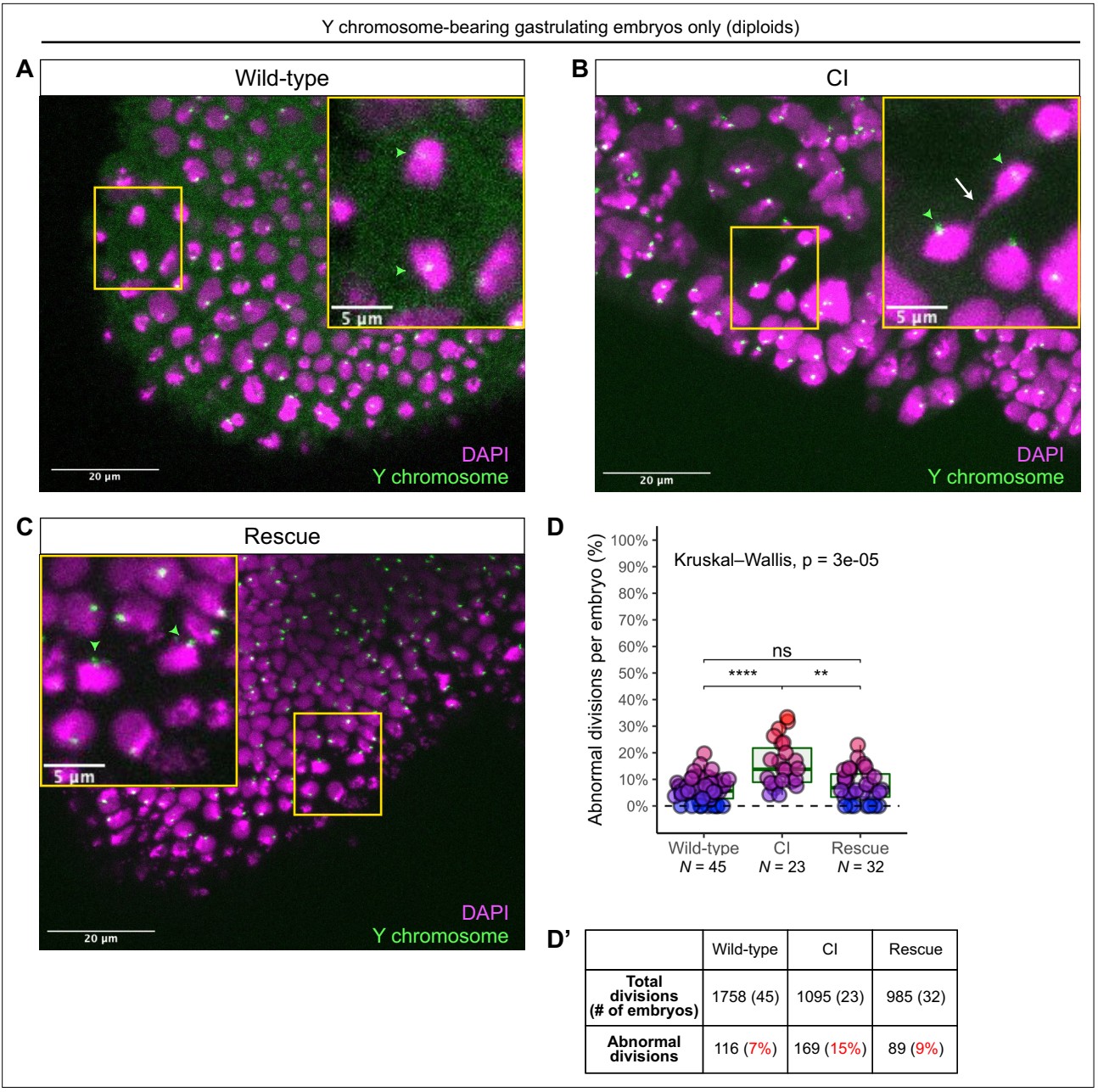

**Figure 6.** Diploid cytoplasmic incompatibility (CI)-derived gastrulating embryos that have escaped the first division defect exhibit increased chromosome segregation errors. Gastrulating embryos from wild-type (**A**), CI (**B**), and rescue (**C**) crosses. Embryos are hybridized with probes that specifically recognize the *D. simulans* Y chromosome (green arrowheads) to select for diploidy (both maternal and paternal chromosome sets present) and counterstained with DAPI (magenta). Yellow boxes indicate zoomed in regions. Scale bars are 20 and 5 μm for unzoomed and zoomed images, respectively. (**A**) Diploid wild-type-derived embryos exhibit relatively normal chromosome segregation. (**B**) Diploid CI-derived embryos have elevated rates of bridging and lagging chromosomes (arrow). (**C**) Diploid rescue-derived embryos exhibit relatively normal chromosome segregation. (**D**) Comparison of the percentage of chromosome segregation errors observed in diploid wild-type-, CI-, and rescue-derived embryos. Each dot represents one embryo. *N* values correspond to the number of embryos scored (biological replicates). Asterisks indicate significance by Mann–Whitney tests. Wild-type to CI: $p = 6.7 \times 10^{-6}$; CI to rescue: $p = 1.1 \times 10^{-3}$. ns = 0.26. (**D'**) Summary of chromosome segregation errors in wild-type-, CI-, and rescue-derived embryos. See also *Figure 6—figure supplements 1 and 2*.

The online version of this article includes the following figure supplement(s) for figure 6:

**Figure supplement 1.** Hatched eggs from cytoplasmic incompatibility (CI) crosses exhibit significantly increased lethality prior to eclosion than those from wild-type or rescue crosses.

**Figure supplement 2.** Non-Y chromosome-containing gastrulating embryos exhibit extensive chromosome segregation errorss.

In accord with previous studies in *D. simulans* (*Callaini et al., 1997*; *Callaini et al., 1996*; *Lassy and Karr, 1996*), we found that there is a second late embryonic lethality associated with CI-derived embryos: between one-fourth and one-third of embryos die after cellularization but before hatching (*Figure 1*). However, while we found embryos derived from CI crosses were relatively normal during nuclear cycles 2–9 (*Figure 2*), previous reports (*Callaini et al., 1996*; *Lassy and Karr, 1996*) observed pre-blastoderm embryos with substantial mitotic defects. In these experiments, the percentage of abnormal CI-derived embryos fixed after a 6 hr age was similar to that of CI-derived embryos fixed immediately after collection, suggesting pre-blastoderm defects manifest from first division defects (*Lassy and Karr, 1996*). Therefore, embryos initially undergoing extensive first division defects may develop slowly through pre-blastoderm divisions if they are aged. As we fixed embryos directly after collecting with no aging time, a possible explanation for the low percentage of abnormal pre-blastoderm embryos we observed is that there was not enough time for embryos experiencing substantial first division segregation errors to develop through the pre-blastoderm stages. Instead, our fixing mostly captured the embryos that correspond to the previously reported percentage (~44%) of the CI-derived embryos that develop normally through the pre-blastoderm stage in spite of a later lethal phase (*Lassy and Karr, 1996*).

As the CI-derived embryos subsequently progressed through the cortical divisions (cycles 10–14), they begin to experience increasingly more severe defects. These late embryonic defects include lagging anaphase chromosomes and chromosome bridging, which directly result in nuclear fallout, and further chromosome bridging during gastrulation (*Figures 3 and 4*, *Figure 4—figure supplement 1*). The normal progression of embryos through cycles 2–9 suggests paternal chromosome segregation was not partially defective in the first division. This is because paternal chromosome bridging in the first division would produce aneuploid daughter nuclei bearing chromosome fragments lacking telomeres. The lack of telomeres would result in detectable breakage–fusion–bridge cycles and amplifications of the aneuploidy in subsequent divisions (*McClintock, 1941*; *Titen and Golic, 2008*), which we did not observe. Our sequencing analysis of cellularized embryos (*Figure 5*) confirms that late-stage CI embryos did not experience partial chromosome loss during the first division. Thus, the mitotic defects first observed in cortical syncytial divisions are unlikely a consequence of CI-induced segmental aneuploidy following the first nuclear cycle.

Consequently, our single embryo PCR and whole genome sequencing of cellularized embryos containing a paternally derived *egfp* transgene revealed late-stage CI embryos were either haploid or diploid (*Figure 5*). The percentage of diploid embryos closely corresponded to the percentage of embryos hatched, suggesting late embryonic lethality is associated with CI-induced haploidy in *D. simulans*, as previously reported (*Callaini et al., 1997*). Importantly, this experiment also demonstrated that the detection of paternally derived chromosomes in that late-stage CI embryos could be used to distinguish between embryos that had experienced first division defects (haploid = only maternal chromosomes, no paternal chromosomes) and embryos that had not experienced any first division defects (diploid = both maternal and paternal chromosomes). As discussed below, this allowed us to uncouple CI-induced late embryo defects from first division defects.

In spite of the strong association between haploidy and lethality, first division-induced haploidy in and of itself cannot explain the defects we observed in CI-derived blastoderm and gastrulating embryos. This is because (1) haploidy is not intrinsically harmful to mitotic divisions in *Drosophila*. For example, in some *Drosophila* mutations that induce haploidy, chromosome segregation occurs normally during cortical divisions (*Tang et al., 2017*). Additionally, meiosis II—essentially a mitotic division of a haploid nucleus—is highly accurate by necessity. Furthermore, (2) any downstream effects of haploidy—such as changes to zygotic gene copy number or loss of zygotic heterozygosity—cannot explain defects first detected in syncytial cortical divisions (cycles 10–13), which do not require zygotic transcription (*Yuan et al., 2016*). In contrast, the observed defects in CI-derived late embryos are more likely due to a second temporally deferred CI-induced defect.

In support of this hypothesis, our observation of increased chromosome segregation errors in diploid CI gastrulating embryos bearing paternally derived Y chromosomes establishes that the defects observed in late-stage CI embryos are not limited to haploids (*Figure 6*). Instead, defects are present in diploid late-stage embryos. Significantly, as discussed above for the paternally derived *egfp* transgene, detection of the Y chromosome by FISH allowed us to select late-stage diploid embryos that had 'escaped' first division defects and instead continued development with both paternal and

maternal chromosome complements. Therefore, the significant increase in mitotic errors observed in diploid CI-derived embryos relative to wild-type-derived embryos demonstrates the existence of a second, CI-induced defect, temporally and possibly mechanistically distinct from the first division defect (*Figure 7A, B*). Significantly, maternally supplied *Wolbachia* independently rescues this later second defect as well (*Figure 7C*).

Interestingly, we also observed several non-Y-bearing gastrulating embryos from CI crosses that had extensive chromosome segregation errors beyond what we had observed for the diploid Y-bearing gastrulating embryos (*Figure 6—figure supplement 2*). Non-Y-bearing embryos may either be diploid (XX) or haploid (XØ). If these embryos were haploid, this observation would suggest that CI could affect both the maternal chromosomes and paternal chromosomes.

One intriguing aspect of the second CI-induced defect is that the embryos progress normally through the early mitotic cycles and then begin to exhibit mitotic defects in the blastoderm stage. The explanation is likely a consequence of the dramatic structural and regulator cell cycle modifications that occur when the dividing nuclei arrive at the cortex (*Farrell and O'Farrell, 2014*). These include heterochromatin formation, initiation of late origins of replication, slowing of DNA replication, activation of zygotic transcription, and metaphase furrow formation (*Farrell and O'Farrell, 2014*; *Li et al., 2014*; *Riggs et al., 2003*; *Seller et al., 2019*; *Seller and O'Farrell, 2018*). The phenotype of numerous maternal-effect mutations that either rely on or disrupt these processes is strikingly similar to the defects observed in CI embryos: normal early pre-cortical divisions followed by extensive mitotic errors and nuclear fallout during the cortical blastoderm divisions (*Sullivan and Theurkauf, 1995*). For example, because of the slowing of DNA replication during the cortical divisions, activation of the S-phase checkpoint is specifically required during this stage. Maternal-effect mutants that disrupt this checkpoint progress normally through the early divisions but exhibit anaphase bridging and nuclear fallout during the late cortical blastoderm divisions as a result of entering metaphase with incompletely replicated chromosomes (*Fogarty et al., 1997*; *Fogarty et al., 1994*). Given the similarity of this phenotype, both in timing and chromosome dynamics, CI-induced late division defects may be due to improper or abnormally slow chromosome replication. Additionally, defects in other events specific to the cortical blastoderm cycles, may also contribute directly or indirectly to CI-induced defects. For example, studies of hybrid incompatibilities between *D. simulans* and *D. melanogaster* show that heterochromatin establishment may be particularly sensitive, and its disruption can result in defects strikingly similar to the late CI defects (*Ferree and Barbash, 2009*). Other important processes, such as those involved in DNA integrity, protein turnover, and cell cycle timing, may be also involved (*Momtaz et al., 2020*).

In considering the mechanism by which paternal *Wolbachia* may induce these defects, the observation that CI-derived blastoderm embryos progress normally through pre-cortical divisions must be noted. One potential explanation is that the chromosomes in CI-derived embryos could be epigenetically marked by *Wolbachia* in the paternal germline. This mark would then persist through the pre-cortical divisions and become disruptive during blastoderm divisions. Interestingly, *Wolbachia* infection results in altered DNA methylation levels in certain wasps, mosquitos, leafhoppers, and *Drosophila* (*LePage et al., 2014*; *Negri et al., 2009*; *Wu et al., 2020*; *Ye et al., 2013*). Should *Wolbachia* in the male germline similarly change the low naturally occurring methylation levels in *D. simulans* (*Deshmukh et al., 2018*), the altered mark may become disruptive in blastoderm divisions, potentially by distorting heterochromatin establishment. However, DNA methylation does not appear to contribute to CI levels (*LePage et al., 2014*).

An alternative explanation for the specificity of the late blastoderm defects comes from studies into hybrid dysgenesis in *D. melanogaster* in which unregulated mobilization of transposable elements results in a spectrum of genetic and developmental defects in the germlines of dysgenic progeny (*Kidwell et al., 1977*). Transposition in progeny can be suppressed when maternally supplied small RNAs mediate silencing of the transposable element (*Czech and Hannon, 2016*). This silencing is associated with increased H3K9 methylation, increased heterochromatin levels, and altered splicing (*Le Thomas et al., 2013*; *Sienski et al., 2012*; *Teixeira et al., 2017*). Given small RNAs can affect chromosomes in trans (*Hermant et al., 2015*), CI may induce a similar small RNA pathway that could epigenetically alter both paternal and maternal chromosomes prior to the first division. As the blastoderm divisions do not require zygotic transcription (*Yuan et al., 2016*), it is unlikely an epigenetic alteration, if occurring, would cause defects via disrupted transcription. Instead, as discussed above,

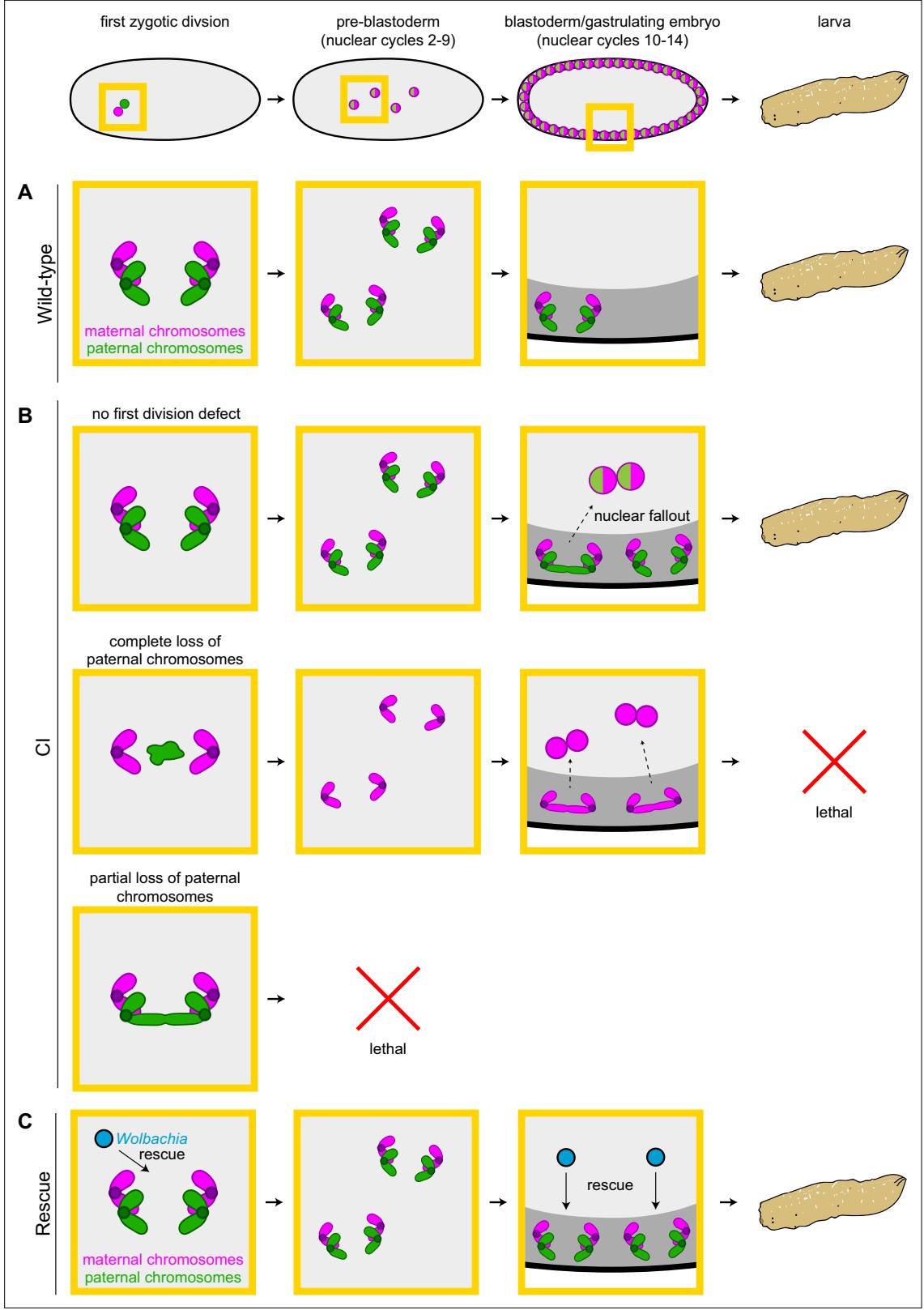

**Figure 7.** Cytoplasmic incompatibility (CI) induces independent first division and mid-blastula transition chromosome segregation errors. (**A**) During the first zygotic division in wild-type derived embryos, paternal (green) and maternal (magenta) chromosomes segregate normally. Chromosome segregation occurs normally during pre-blastoderm, blastula, and post-cellularization divisions. Embryos hatch. (**B**, top row) In CI-derived embryos, if there are no segregation defects during the first division, embryos develop as diploids containing full maternal and paternal chromosome sets. Pre-

*Figure 7 continued on next page*

*Figure 7 continued*

blastoderm divisions proceed normally. However, during blastoderm stages, CI induces a second set of defects, which cause chromosome segregation errors and subsequent nuclear fallout (dashed arrow). Chromosome segregation errors continue during gastrulation. These defects occur at moderate frequencies and embryos hatch. (**B**, middle row) If the paternal chromosomes are completely excluded during the first division, embryos develop as haploids from only the maternal chromosome set. Pre-blastoderm divisions proceed normally, followed by increased chromosome segregation errors and nuclear fallout during blastoderm divisions. Chromosome segregation errors continue during gastrulation. Perhaps due to CI being strong in haploid embryos (*Bonneau et al., 2018*), this second set of CI-induced defects is more severe, and embryos fail to hatch, due to their haploidy. (**B**, bottom row) If the paternal chromosomes are partially lost during the first divisions, embryos arrest due to severe aneuploidy. (**C**) Maternally supplied *Wolbachia* (blue circles) rescue both the first division defects and the late-stage defects independently.

an epigenetic change may disrupt key aspects of the mid-blastula transition, which in turn could result in the observed errors.

Insight into the molecular mechanism of CI came with the discovery of the *Wolbachia*-encoded Cifs that play a key role in CI and rescue (*Beckmann et al., 2017*; *LePage et al., 2017*). A combination of molecular, genetic, and biochemical studies provided compelling evidence that the *Wolbachia* encoded genes, *CidB* and *CidA*, act as a paternally supplied toxin and maternally supplied anti-toxin, respectively (*Beckmann et al., 2017*; *Horard et al., 2022*; *Wang et al., 2022*). However, a toxin/anti-toxin model for CI does not easily explain the cortical blastoderm defects occurring after many rounds of normal mitotic cycles. This is because a paternally supplied toxin is expected to be diluted with every round of division, and therefore its induced defects would decrease over time. Similarly, a second set of CI/rescue elements, *CinA* and *CinB*, is also proposed to act in a toxin/anti-toxin manner (*Chen et al., 2019*; *Sun et al., 2022*). An alternative possibility is that Cifs may epigenetically modify paternal and maternal chromosomes to mediate CI and rescue (*Kaur et al., 2022*).

How and if these proteins also contribute to CI-induced late-embryo defects remains to be determined. It is possible that early and late defects are caused by common acute mechanisms involving the known Cifs. This may be due to an induced chromosomal change that produces defects at different times given natural variation in developmental processes among a population of embryos. In contrast, chromosome segregation errors in diploid embryos that have progressed normally through the first division may suggest minimal Cif activity and an additional set of *Wolbachia* genes that induce late-embryo defects. Unlike the first division errors, CI-induced mitotic defects in late embryos do not appear to result from abnormal condensation, alignment, or timing of metaphase exit (*Figure 4*). Instead, the observed chromosome bridging is strikingly similar to embryos exposed to the DNA replication inhibitor aphidicolin (*Farrell et al., 2012*; *Fasulo et al., 2012*), suggesting CI-derived blastoderm embryos may be entering anaphase with incompletely replicated chromosomes. These differences could suggest a separate proximate cause for early and late defects, or provide additional insight into Cif-induced defects. Thus, any model of CI and rescue, be it toxin/anti-toxin, lock-key, titration, or timing, must account for the fact that some effects of *Wolbachia* on the sperm are not realized until hours and many cell cycles later when the embryos progress through the late blastoderm divisions and the mid-blastula transition.

## Materials and methods

**Key resources table**

| Reagent type (species) or resource | Designation | Source or reference | Identifiers | Additional information |
|---|---|---|---|---|
| Strain, strain background (*Wolbachia riverside*) | *Wolbachia (wRi)* | *O'Neill and Karr, 1990* | | |
| Genetic reagent (*Drosophila simulans*) | Infected | *Serbus et al., 2015* | | Infected with *wRi* |
| Genetic reagent (*Drosophila simulans*) | Uninfected | *Casper-Lindley et al., 2011* | | Generated by tetracycline-curing *wRi*-infected stocks |
| Genetic reagent (*Drosophila simulans*) | *egfp* uninfected | *Holtzman et al., 2010* | NDSS: 14021-0251.275 | Obtained from the National *Drosophila* Species Stock Center (NDSS) |

*Continued on next page*

*Continued*

| Reagent type (species) or resource | Designation | Source or reference | Identifiers | Additional information |
|---|---|---|---|---|
| Genetic reagent (*Drosophila simulans*) | *egfp Wolbachia*-infected | This paper | | Generated by crossing *Wolbachia* from infected stock into the *egfp* uninfected stock |
| Sequence-based reagent | *egfp* 5' | Cruachem | PCR primers | ATCAAGCTTGTGA GCAAGGGCGAGGAGC |
| Sequence-based reagent | *egfp* 3' | Cruachem | PCR primers | ACCTCGAGCTACTT GTACAGCTCGTCCATGC |
| Sequence-based reagent | Y-chr-488 | *Ferree and Barbash, 2009* | FISH probe | (AAT-AAA-C)$_4$ conjugated to Alexa488, synthesized by Integrated DNA technologies |
| Peptide, recombinant protein | Rhodamine-labeled histone | This paper | | Recombinant rhodamine-histone fusion protein |
| Antibody | Anti-centrosomin (Rabbit polyclonal) | *Megraw et al., 1999* | | (1:200) |
| Antibody | Anti-rabbit-488 (Goat polyclonal) | Thermo Fisher | Cat# A-11008 | (1:1000) |
| Gene (*Drosophila simulans*) | *D. simulans* reference genome | *Clark et al., 2007* | WUGSC mosaic 1.0/droSim1 assembly | UCSC Genome Browser |
| Gene (*Wolbachia riverside*) | *wRi* reference genome | *Klasson et al., 2009* | CP001391.1 | GenBank |
| Gene (*p-egfp* plasmid) | *egfp* | Addgene | | |
| Software, algorithm | BWA-MEM2 | *Md et al., 2019* | | Version 2.2.1 |
| Software, algorithm | Picard tools | Broad Institute | | Version 2.27.1 |
| Software, algorithm | BEDTools | *Quinlan and Hall, 2010* | | Version 2.26.0 |
| Software, algorithm | Leica Application Suite Advanced Fluorescence | Leica | | |
| Software, algorithm | R | R Core Team | | Version 4.0.5 |
| Software, algorithm | ggplot2 | *Wickham, 2016* | | |
| Other | DAPI in Vectashield | Vector Laboratories | Cat# H-1200 | DNA stain (1.5 µg/ml) in mounting media |
| Other | Halocarbon oil | Millipore Sigma | Cat# H8898 | For mounting embryos for live imaging |

### *Drosophila* stocks

All stocks were grown on standard brown food (*Sullivan et al., 2000*) at 25°C with a 12 hr light/dark cycle. Uninfected *D. simulans* stocks were generated by tetracycline-curing a *w*- *Wolbachia* (*wRi*)-infected stock (*Casper-Lindley et al., 2011*; *Serbus et al., 2015*). Uninfected and infected stocks were allowed to grow for many generations prior to their use. Throughout these experiments, we routinely checked for the presence/absence of *Wolbachia* by PCR with primers against the 16s rRNA gene of *Wolbachia*.

An uninfected *egfp*-bearing stock was obtained from the National *Drosophila* Species Stock Center (Cornell College of Agriculture and Life Sciences; #275; *w*[501]; PBac(GreenEye.UAS.tubEGFP)Dsim3) (*Holtzman et al., 2010*). *Wolbachia* was introduced to this stock by crossing to *Wolbachia*-infected females. Progeny was backcrossed to obtain flies homozygous for *egfp*. Stocks were routinely checked for *Wolbachia* and *egfp* presence by PCR with primers against the 16s rRNA gene of *Wolbachia* and *egfp*, respectively. Males from this stock were used for experiments in which infected father transmitted an *egfp* transgene to offspring.

Embryos were collected from crosses of 3- to 5-day-old flies (*Figures 1–3*, *Figure 1—figure supplement 1*) or 2- to 4-day-old flies (*Figures 4–6*, *Figure 4—figure supplement 1*). For experiments in *Figures 1–3* embryos were collected for 4 days after the initial collection. For all other experiments, embryos were collected only on the initial collection.

## Egg hatch assays

For experiments involving egg hatch assays (*Figures 1 and 4*, *Figure 1—figure supplement 1*), collected embryos were aged in a humid chamber at 25°C for at least 30 hr before hatched eggs were counted.

## Embryo fixation

For fixed experiments assaying embryo stage, abnormalities, and nuclear fallout, 1–6 hr (*Figure 1A–B'*), 2.5–3 hr (*Figure 1C–D'*), 0–4 hr (*Figure 2*), and 1–4 hr (*Figure 3*) embryos were dechorionated in bleach, washed thoroughly in water, and transferred to a 1:1 ratio of heptane and 32% paraformaldehyde (Electron Microscopy Sciences 15714) for 5 min. Paraformaldehyde was subsequently removed and replaced with methanol and shaken vigorously. Heptane was removed and embryos stored in methanol at 4°C. Embryos were mounted directly in PI (*Figure 1A–B'*) or DAPI with Vectashield (Vector H-1200) (*Figures 1C–D'–3*).

For fixed experiments analyzing nuclear detachment from centrosomes (*Figure 3*), 1–4 hr embryos were initially fixed as described above. Embryos were rehydrated in PBT (phosphate-buffered saline [PBS] + 0.05% Triton + 1% bovine serum albumin [BSA]), blocked for 1 hr, and incubated with rabbit anti-centrosomin antibody (1:200) (*Megraw et al., 1999*). After 3× washes in PBT, embryos were incubated with anti-rabbit-Alexa488 secondary (1:1000 Thermo Fisher A-11008). Embryos were washed 3× in PBT, rinsed 4× in PBS, and counterstained with DAPI in Vectashield (Vector Laboratories H-1200-10).

For fixed experiments assaying chromosome segregation errors in gastrulating embryos (*Figure 4—figure supplement 1*), 3–5 hr embryos were dechorionated in bleach, washed thoroughly in water, and permeabilized in heptane for 2.5 min. Embryos were fixed by adding an equal volume methanol to the heptane and shaking vigorously. Heptane was removed, and embryos stored at 4°C in methanol. Embryos were mounted directly in DAPI with Vectashield.

For fixed experiments involving FISH (*Figure 6*, *Figure 6—figure supplement 2*), 2–5 hr embryos were dechorionated in bleach, washed thoroughly in water, and permeabilized in ice-cold heptane for 2.5 min. Embryos were fixed in an ice-cold 4% paraformaldehyde–46% PBS–50% heptane mixture for 10 min. Following removal of the paraformaldehyde–PBS solution, an equal volume of ice-cold methanol was added to the heptane and shaken vigorously. Heptane was then removed. Embryos were then stored at 4°C in methanol.

## Live embryo staging

For experiments involving live embryo staging (*Figure 1—figure supplement 1*, *Figure 5*), embryos were collected for 45 min, hand dechorionated, covered in halocarbon oil, and aged in a humid chamber at 25°C for 2.5 hr. Embryo stage was either scored after this time (*Figure 1—figure supplement 1*) or for every 60 min (*Figure 5*). Embryos were staged using an Olympus SZH10 high-powered dissecting scope. Live images presented in *Figure 1—figure supplement 1* were acquired on a Zeiss Axiozoom V.16 microscope equipped with a Zeiss AxioCam HRm monochrome camera. Images were acquired with Zeiss Zen software.

## Embryo injection

For live imaging experiments (*Figure 4*), 0.5–1.5 hr embryos were hand dechorionated, placed in halocarbon oil, and injected with rhodamine-labeled histone. Embryos were imaged directly after injection in areas adjacent to the injection site.

## Fluorescence in situ hybridization

Alexa488-conjugated probes targeting the *D. simulans* Y-chromosome (AAT-AAA-C)$_4$ (*Ferree and Barbash, 2009*) were synthesized by Integrated DNA Technologies (Coralville, IA, USA). Paraformaldehyde-fixed embryos were rehydrated in PBT (PBS + 0.05% Triton + 1% BSA). Embryos were washed in 4× saline sodium citrate (SSC), 10% formamide, 50 mM imidazole for 1 hr at 37°C. Embryos were hybridized with probes in hybridization buffer (4× SSC, 10% formamide, 0.0001% dextran sulfate) at 92°C for 3 min then 37°C overnight. Embryos were washed 3× in 2× SSC, 50% formamide for 10 min at 37°C, rinsed 4× in PBS, and counterstained with DAPI in Vectashield.

## Confocal imaging

Live and fixed embryo imaging was performed on an inverted Leica DMI6000 SP5 scanning confocal microscope. DAPI was excited with a 405 nm laser and collected from 410 to 480 nm. Alexa488 was excited with a 488 nm laser and collected from 518 to 584 nm. Rhodamine was excited with a 543 nm laser and collected from 555 to 620 nm. PI was excited with 514 and 543 nm lasers and collected from 627 to 732 nm. Embryos were imaged with either ×10/0.3, ×20/0.75, ×40/1.25 oil, or ×63/1.4 oil objectives. All imaging was performed at room temperature. Images were acquired with Leica Application Suite Advanced Fluorescence software. For live imaging experiments (*Figure 4*), timepoints between images were every 12–60 s depending on the size of the z-stack.

## Single embryo PCR analysis

Cellularized blastoderms were individually squashed and then lysed in 10 µl buffer containing Proteinase K and ThermoPol reaction buffer (New England BioLabs) for 45 min at 60°C then 10 min at 95°C. PCR was run with 1 µl sample in 20 µl total reaction volume, using primers targeting *egfp* (5′: ATCAAGCTTGTGAGCAAGGGCGAGGAGC; and 3′: ACCTCGAGCTACTTGTACAGCTCGTCCATGC) (Cruachem). PCR was run as: 10 min at 95°C, 31× (30 s at 95°C, 1 min at 60°C, 1 min at 72°C), 10 min at 72°C. PCR products were resolved on a 1% agarose gel. These conditions regularly produced an ~1.4 kb band only when the *egfp* transgene was present.

## Single embryo sequencing

Cellularized blastoderms were individually squashed, frozen in liquid nitrogen, and stored at −80°C. Library preparation (NexteraXT kit) and paired-end sequencing (Illumina HiSeq, 2 × 150 bp) was performed by Azenta Life Sciences (Indianapolis, IN, USA). As samples contained host DNA, *Wolbachia* (*wRi*) DNA, and an *egfp* insertion, we assembled a reference genome consisting of the *D. simulans* genome (WUGSC mosaic 1.0/droSim1 assembly, *Clark et al., 2007*, UCSC Genome Browser, Santa Cruz, CA, USA), a *wRi* genome (*Klasson et al., 2009*, GenBank CP001391.1), and the *egfp* sequence from the *p-egfp* plasmid (Addgene, Watertown, MA, USA). We additionally included a 714-bp randomized sequence as a negative control.

Reads were aligned to the reference genome using BWA-MEM2 (2.2.1) (*Md et al., 2019*). Duplicate reads were removed using Picard tools (Broad Institute, 2018, Picard Tools, 2.27.1, http://broadinstitute.github.io/picard/) and low-quality reads ($q < 20$) were subsequently removed. BEDTools (2.26.0) (*Quinlan and Hall, 2010*) was used to assign depth of coverage at each position in the genome. Read alignment and processing were performed using the Hummingbird Computational Cluster (UC Santa Cruz, Santa Cruz, CA, USA). Gene coordinate positions were determined in the UCSC Genome Browser.

Percent depth of a gene was calculated by dividing the average depth across a gene by the average depth across the whole genome for that embryo and multiplying by 100%. Embryos were considered diploid if the mean depth of reads aligning to the *egfp* transgene was meaningful (around 50% for heterozygote embryos) and reads were distributed evenly across the entirety of the *egfp* transgene. To decrease stochastic noise and accurately assess potential chromosome loss, we analyzed five genes from each chromosome/chromosome arm (Y, X, 2L, 2R, 3L, 3R, 4). Chromosome/chromosome arm loss was considered if the depth of reads across multiple genes on a chromosome/chromosome arm dropped from either 100% to 50% (diploid) or from 100% to 0% (haploid). As a proof of concept, an example of natural chromosome 'loss' can be observed in male embryos (Y-linked genes present) in which the depth of reads mapping to genes on the X chromosome are ~50% of the mean genome depth (hemizygous).

## Egg-to-adult assays

Egg hatch assays were performed using embryos collected from 2- to 4-day-old flies. Eggs were counted and transferred to a new collection plate in a new collection bottle. Hatched eggs were counted after at least 30 hr. Adults were counted for each plate for as long as new adults were eclosing.

## Statistical analyses

Independent runs of an experiment were considered technical replicates. For each experiment, different cells (for scoring mitotic abnormalities directly preceding nuclear fallout), embryos (for

scoring abnormalities, chromosome segregation errors, *egfp* presence/absence, and sequencing), or crosses (for scoring egg hatch and egg-to-adult rates) were considered biological replicates. Specific sample sizes were not explicitly determined prior to experimentation. Instead, each experiment was performed independently at least three times (i.e., three technical replicates), with the exception of the sequencing experiment (*Figure 5D, E*, *Figure 5—figure supplement 1*) in which embryos were collected once and sequenced in two batches. Additionally, the egg hatch assay for *Figure 1—figure supplement 1* was performed once. Independently collected data for embryo staging in *Figure 1A* were pooled. No data (e.g., outliers) were excluded from analyses. Experiments were analyzed unblinded.

All statistical analyses were performed in R (4.0.5, R Core Team). The following statistical tests were used: $\chi$-squared tests (*Figure 1*, *Figure 1—figure supplement 1*) were performed on pooled data, as the null hypothesis is the frequency of embryos reaching the blastoderm stage should equal the frequency of embryos hatching; $\chi$-squared tests (*Figure 4*), as the null hypothesis is the frequency of chromosome segregation errors should be similar between 'fallout' and 'non-fallout' nuclei; two-sided paired *t*-test (*Figure 1*), as samples for hatch analysis and embryo staging were collected in pairs and data were normally distributed as determined by Shapiro–Wilk tests (wild-type hatch p = 0.6; wild-type blastulation p = 0.4, CI hatch p = 0.8, CI blastulation p = 0.2); two-sided Fisher's exact test (*Figure 2*), as multiple percentages were compared with the null hypothesis that frequencies should not be different across different genotypes and/or developmental stages; Kruskal–Wallis test (*Figures 3 and 6*, *Figure 4—figure supplement 1*, *Figure 6—figure supplement 1*), as more than two distributions were compared with no assumption of normalcy; and Mann–Whitney tests (*Figures 3 and 6*, *Figure 4—figure supplement 1*, *Figure 6—figure supplement 1*), as two distributions were compared with no assumption of normalcy.

## Figure preparation

Graphs were created in R using the ggplot2 package (*Wickham, 2016*). To improve the clarity of certain panels, images were adjusted for brightness and contrast in FIJI. Figures were assembled in Adobe Illustrator (Adobe, San Jose, CA, USA).

## Acknowledgements

We thank Dr. Benjamin Abrams (UCSC Life Sciences Microscopy Center, RRID: SCR_021135) for his technical support and assistance with microscopy experiments. We thank Dr. Shelbi Russell and Dr. Andreas Rechsteiner for their helpful advice on sequencing experiments and analysis. We thank Dr. Timothy Megraw who provided our lab with the anti-centrosomin antibody.

## Additional information

### Funding

| Funder | Grant reference number | Author |
|---|---|---|
| National Institutes of Health | NIGMS-1R35GM139595 | William Sullivan |

The funders had no role in study design, data collection, and interpretation, or the decision to submit the work for publication.

### Author contributions

Brandt Warecki, Conceptualization, Data curation, Software, Formal analysis, Supervision, Validation, Investigation, Visualization, Methodology, Writing – original draft, Writing – review and editing; Simon William Abraham Titen, Conceptualization, Formal analysis, Validation, Investigation, Methodology, Writing – review and editing; Mohammad Shahriyar Alam, Conceptualization, Formal analysis, Investigation, Writing – original draft, Writing – review and editing; Giovanni Vega, Formal analysis, Validation, Investigation; Nassim Lemseffer, Formal analysis, Investigation, Methodology; Karen Hug, Formal analysis, Investigation; Jonathan S Minden, Methodology; William Sullivan, Conceptualization,

Formal analysis, Supervision, Funding acquisition, Validation, Investigation, Methodology, Writing – original draft, Project administration, Writing – review and editing

### Author ORCIDs
Brandt Warecki http://orcid.org/0000-0002-8025-6246
Mohammad Shahriyar Alam http://orcid.org/0000-0003-2943-8084
William Sullivan http://orcid.org/0000-0002-1756-4174

### Decision letter and Author response
Decision letter https://doi.org/10.7554/eLife.81292.sa1
Author response https://doi.org/10.7554/eLife.81292.sa2

## Additional files

### Supplementary files
• MDAR checklist

### Data availability
Sequencing data have been deposited in the Sequence Read Archive (BioProject PRJNA848235). Figure 5—source data 1 contains the raw unedited gel used to generate Figure 5B as well as Figure 5 prepared with the raw gel. Figure 5—source data 2 contains numerical source data to generate Figure 5E. All reagents used are available upon request.

The following dataset was generated:

| Author(s) | Year | Dataset title | Dataset URL | Database and Identifier |
|---|---|---|---|---|
| Warecki B, Titen S, Alam MS, Vega G, Lemseffer N, Hug K, Minden JS, Sullivan W | 2022 | *Drosophila simulans* cellularized blastoderm embryos | http://www.ncbi.nlm.nih.gov/bioproject/?term=PRJNA848235 | NCBI BioProject, PRJNA848235 |

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
