## [Editor Report]

This manuscript investigates the cellular and developmental defects underlying Wolbachia-induced cytoplasmic incompatibility (CI), which occurs when male insects harboring the endosymbiont bacteria Wolbachia fertilize eggs of uninfected females, triggering embryonic lethality. Previous work showed CI-induced defects early in embryogenesis (at first mitosis), whereas this work provides the detailed characterization of later defects including loss of nuclei ("nuclear fallout"), determination of haploidy versus diploidy in the bacteria-mediated lethality, and evidence that the mechanism of late embryonic defects is independent of the ones that drive early embryonic defects. The strength of evidence provided is compelling, including beautiful single embryo PCR analysis, and convincing light microscopy. This is a technically superb set of experiments. The significance of the findings is somewhat modest due to it being an incremental step forward, but it will be useful to those in the field.

---

## [Decision Letter]

**Decision letter after peer review:**

Thank you for submitting your article "*Wolbachia* action in the sperm produces developmentally deferred chromosome segregation defects during the *Drosophila* mid-blastula transition" for consideration by *eLife*. Your article has been reviewed by 3 peer reviewers, including Chris Q Doe as Reviewing Editor and Reviewer #1, and the evaluation has been overseen by Marianne Bronner as the Senior Editor. The following individual involved in the review of your submission has agreed to reveal their identity: Timothy Karr (Reviewer #2).

Essential revisions:

1. The authors cite several examples from other insects showing late developmental defects in embryos derived from CI crosses (Bonneau et al., 2018; Callaini et al., 1997; Callaini et al., 1996; Duron and Weill, 2006). The authors should more clearly mention what is novel about their study compared to previous studies documenting late embryonic defects.

2. Do you think that it might be worth staining for the Wolbachia CidA/B or CinA/B proteins? Perhaps they find a way to persist until MBT and then become activated by a zygotic maternal factor? If their localization during fertilization to gastrulation is known, it would be good to assay it or at least discuss it.

3. Beginning line 14, "fail to develop due to the immediate action of Wolbachia-produced factors…" suggests that Wolbachia "factors" are in sperm, but the alternative hypothesis that Wolbachia effect host proteins during spermatogenesis that are responsible for CI cannot be strictly eliminated. Suggest re-wording.

4. Line 22- "Thus, Wolbachia in the sperm…" is a statement in conflict with previously published work that demonstrated the absence of Wolbachia in sperm Snook, R. R., et al., Genetics 155, 167 178 (2000). Suggest re-wording.

5. Line 38, while certainly a "conditional sterility" phenotype, should stress that this is also a unique "paternal effect", an important distinction since there are many hundreds of gene mutations that cause male sterility that have nothing directly to do with sperm function per se.

6. In the paragraph beginning line 96, it is unclear how the authors distinguish the early and later phenotypes mechanistically, have they excluded the possibility that variation in the severity of the early defects, which may be difficult to observe much less quantify, could lead to the manifestation at later stages? The answer to this question is important in assessing the veracity of the last sentence in the paragraph that clearly states the independence of these two phases of Wolbachia-induced defects during embryonic development.

7. Having looked at many thousands of developing *Drosophila* embryos over many years, it is a common (albeit infrequent) observation of mitotic defects in normal wild-type embryos. These observations and this reviewer's experience make the claim of "0%" suspect. Can the authors put this number in a larger perspective?

8. Section line 326, this section was a bit confusing as it relates to Figure 5E. Does the grid plot of coverage indicate just the genes that were measured and are used as representative of the entire chromosome? Can they say with certainty that all genes are present as expected for zero chromosome loss (ie, very small regions?). There appears to be some "noise" in the data as there is a mixture of green and black? The authors appear to allude to this in pp beginning line 352.

9. Line 395, Figure 6/Figure S4 should/could be explained a bit clearer? This very short paragraph and reference to S4 could be expanded for clarity. For example, is it the case that the 2% of the 966 eggs scored led to only ~70% that led to viable adults? Please clarify further as this result is both novel and important.

10. Line 418, there is a troubling lack of discussion about the apparent striking difference in CI phenotypes reported in comparison with those of Callaini et al. and Lassy and Karr. Both laboratories reported extensive early pre-blastoderm mitotic defects. This is dealt with in a rather cavalier manner with the sentence, "In contrast with previous reports (Lassy and Karr, 1996), we found that these embryos initially proceeded normally through nuclear cycles 2-9". It is not clear why the authors did not include the Callaini 1996 paper on this topic, but this discrepancy should be addressed more thoroughly. It should also be noted that Table 1 in the Lassy and Karr paper reported the percentages of CI developmental defects up to 6 hours. They reported ~56% mitotic defects which, by subtraction gives a value for "normal" development at ~44% which is remarkably close to the 40% cited by the authors for embryos that reach the blastoderm stage.

11. The phrase, (line 469) "completely separate from the first division defect" while clearly separate in time, should be mitigated to reflect the more reasonable phrase, "… more likely due to …" as the authors did above (line 458). Clearly, the authors have built a very strong case for these separate phenomena, but probably best to leave out the hyperbole.

12. Revise the abstract to acknowledge the previous literature that embryos can manifest with cytological errors hours into development, including in D. simulans. The authors communicated this situation nicely in the Introduction and can adopt the clarity in the abstract that the key question they intend to address is whether or not the chromosome errors at the 1st mitosis and later (both of which are well documented already) occur due to an initial error in the parental chromatin that carries forward or to independent new types of errors after the first mitosis. This is the question they aim to speak to in the work and it can be more clearly delineated as why it's important, what methods they used, and what the basic results were. For instance, this sentence could be revised from "In CI, embryos from crosses…fail to develop due to the immediate action of Wolbachia-produced factors in the first zygotic division" to "In CI, a paternally-delivered modification of the sperm leads to chromatin defects and lethality during and after the first mitosis of embryonic development in a variety of insect species" (published references: Culex, Aedes, Dmel, Dsim, etc).

13. In the intro, delete the word "sterility" used only in the first paragraph to unify the terminology with embryonic lethality or post-fertilization incompatibility in the paper since many readers may assume sterility refers to a pre-fertilization error that inhibits fertilization. That is not the case here.

14. Line 113: It would help the paper and readers to change the subheader titles to emphasize the new discoveries of this study rather than what's known in previous literature. For instance, instead of stating CI produces late embryo lethality, add the new details of the percentage and type of delays seen here in the subhead.

15. Line 121-122. D. simulans is also a useful model since it was previously shown to have defects after the first mitosis. That is worth noting at this point in the manuscript as it sets the stage for your study.

16. Line 158: Cite Lassy and Karr 1996. -Lassy CW, Karr TL. Cytological analysis of fertilization and early embryonic development in incompatible crosses of *Drosophila* simulans. Mech Dev. 1996;57:47-58. -Callaini G, Riparbelli MG, Giordano R, Dallai R. Mitotic Defects Associated with Cytoplasmic Incompatibility in *Drosophila* simulans. Journal of Invertebrate Pathology. 1996;67:55-64.

---

## [Author Response]

Essential revisions:1. The authors cite several examples from other insects showing late developmental defects in embryos derived from CI crosses (Bonneau et al., 2018; Callaini et al., 1997; Callaini et al., 1996; Duron and Weill, 2006). The authors should more clearly mention what is novel about their study compared to previous studies documenting late embryonic defects.

We have now rewritten the abstract and introduction and added a section to the discussion that highlights what is novel about our work. In brief, the previous studies either (1) only mentioned late embryonic defects in passing, (2) did not determine if late embryonic defects were independent of causes of the initial defects, or (3) interpreted the defects as resulting from haploid development due to the first division defects. Thus, if origin of the late embryonic defects remained poorly understood. In our revisions, we have directly stated this unresolved question in the abstract (lines 15-17), introduction (lines 116-120), and discussion (lines 546-547). We then follow each of these with an explanation for how our work directly addresses this question and adds to the field.

2. Do you think that it might be worth staining for the Wolbachia CidA/B or CinA/B proteins? Perhaps they find a way to persist until MBT and then become activated by a zygotic maternal factor? If their localization during fertilization to gastrulation is known, it would be good to assay it or at least discuss it.

We have initiated these studies and have not found evidence of staining. We are currently pursuing this question with western blots, but we believe this is beyond the scope of this paper.

3. Beginning line 14, "fail to develop due to the immediate action of Wolbachia-produced factors…" suggests that Wolbachia "factors" are in sperm, but the alternative hypothesis that Wolbachia effect host proteins during spermatogenesis that are responsible for CI cannot be strictly eliminated. Suggest re-wording.

We have reworded this sentence to include both possibilities:

“In CI, a paternally-delivered modification of the sperm leads to chromatin defects and lethality during and after the first mitosis of embryonic development in multiple species” (lines 13-15).

4. Line 22- "Thus, Wolbachia in the sperm…" is a statement in conflict with previously published work that demonstrated the absence of Wolbachia in sperm Snook, R. R., et al., Genetics 155, 167 178 (2000). Suggest re-wording.

This is our mistake, and we thank the reviewers for pointing this out. We have changed the sentence as follows:

“Thus, *Wolbachia* action in the sperm induces developmentally deferred defects that are not a consequence of the first division errors” (lines 24-25).

5. Line 38, while certainly a "conditional sterility" phenotype, should stress that this is also a unique "paternal effect", an important distinction since there are many hundreds of gene mutations that cause male sterility that have nothing directly to do with sperm function per se.

We have reworded this sentence as follows:

“CI is a form of *Wolbachia*-induced paternal effect embryonic lethality” (line 40).

6. In the paragraph beginning line 96, it is unclear how the authors distinguish the early and later phenotypes mechanistically, have they excluded the possibility that variation in the severity of the early defects, which may be difficult to observe much less quantify, could lead to the manifestation at later stages? The answer to this question is important in assessing the veracity of the last sentence in the paragraph that clearly states the independence of these two phases of Wolbachia-induced defects during embryonic development.

The key observations in distinguishing the early and late division defects are:

1. A large fraction of the CI-derived embryos develops seemingly normally through nuclear cycles 2-9. This is not expected to be the case if these embryos were aneuploid because of loss of paternal chromosomes in the first division (i.e. mitotic catastrophe)

2. Sequence analysis reveals that ~40% of CI-derived embryos that have reached the cortical divisions are diploid

3. Embryos containing paternally-derived chromatin exhibit chromosome segregation errors during gastrulation

We note that we did not use the term “independent” to specifically mean two separate mechanisms, but simply that late defect does not depend on the initial defect. We appreciate these points were unclear in our original submission. We have now rewritten this paragraph to better clarify how we differentiated early and late embryonic defects as well as reflect potential uncertainty that could be caused by difficult-to-observe/quantify variation in the severity of early defects (lines 131-145).

7. Having looked at many thousands of developing *Drosophila* embryos over many years, it is a common (albeit infrequent) observation of mitotic defects in normal wild-type embryos. These observations and this reviewer's experience make the claim of "0%" suspect. Can the authors put this number in a larger perspective?

For the experiment referenced, we were interested in whether there were gross abnormalities that would be characteristic of large-scale CI-induced defects. As such, we scored embryos as whole units as either normal/abnormal, and thus minor defects might not be counted. This was not clear in our initial submission. We have added the following sentence to clarify:

“As we were interested in the timing with which CI induces large-scale defects throughout development, we scored embryos as whole units, which potentially excludes minor defects” (lines 240-242).

When more sensitive assays are performed (such as scoring individual nuclei or cells), mitotic defects can be observed in wild-type embryos. As the reviewer mentions, we can observe these common but infrequent defects in wild-type embryos when we perform these assays such as in scoring for nuclear fallout (Figure 3) or chromosome segregation errors (Figure 4 —figure supplement 1, Figure 6). We discuss this in lines 278-280.

8. Section line 326, this section was a bit confusing as it relates to Figure 5E. Does the grid plot of coverage indicate just the genes that were measured and are used as representative of the entire chromosome? Can they say with certainty that all genes are present as expected for zero chromosome loss (ie, very small regions?). There appears to be some "noise" in the data as there is a mixture of green and black? The authors appear to allude to this in pp beginning line 352.

Yes, the grid plot indicates just the genes that were measured and are representative of the entire chromosome. We chose genes spread across the major chromosome arms where coverage was reliable based on the wild-type and rescue embryos. We agree there is noise, as we are performing single embryo sequencing, which was at the limit of technology. While we cannot rule out small deletions, we believe such small deletions cannot regularly cause the large-scale defects we observe. We have rewritten this paragraph to clarify how our analysis was performed as well as address the uncertainty of very small deletions we cannot detect due to noise (lines 416-430).

9. Line 395, Figure 6/Figure S4 should/could be explained a bit clearer? This very short paragraph and reference to S4 could be expanded for clarity. For example, is it the case that the 2% of the 966 eggs scored led to only ~70% that led to viable adults? Please clarify further as this result is both novel and important.

We have expanded this paragraph (lines 497-515). In brief, we first performed hatch assays and then determined the number of hatched eggs from these assays that subsequently developed into adults. Thus, as the reviewer mentions for CI, out of 966 eggs collected, 137 hatched and from those 94 developed into adults (69% of hatched eggs). In comparison, for wild-type, out of 548 hatched eggs, 520 (95%) developed into adults. We agree this is a potentially important result suggesting CI can act a remarkably delayed time. In our revised paragraph, we more clearly state the experimental setup and results to clarify this finding.

10. Line 418, there is a troubling lack of discussion about the apparent striking difference in CI phenotypes reported in comparison with those of Callaini et al. and Lassy and Karr. Both laboratories reported extensive early pre-blastoderm mitotic defects. This is dealt with in a rather cavalier manner with the sentence, "In contrast with previous reports (Lassy and Karr, 1996), we found that these embryos initially proceeded normally through nuclear cycles 2-9". It is not clear why the authors did not include the Callaini 1996 paper on this topic, but this discrepancy should be addressed more thoroughly. It should also be noted that Table 1 in the Lassy and Karr paper reported the percentages of CI developmental defects up to 6 hours. They reported ~56% mitotic defects which, by subtraction gives a value for "normal" development at ~44% which is remarkably close to the 40% cited by the authors for embryos that reach the blastoderm stage.

We apologize for this. In our revised manuscript, we have more thoroughly discussed these differences. In the Lassy and Karr paper, the percentage of abnormal embryos fixed after a 6h aging period was similar to that of embryos fixed immediately after collection, suggesting the pre-blastoderm defects observed were likely due to first division defects. In our experiments, we fixed embryos directly after collecting with no aging time. We believe this did not allow enough time for embryos experiencing extensive first division defects to substantially develop through nuclear cycles 2-9. We believe if we were to age the embryos 2-6h after our collections we would see a similar percentage of abnormal 2-9 embryos as Lassy and Karr, since these embryos would now have time to develop past the first division defect. Therefore, as mentioned by the reviewer, the cycle 2-9 embryos we observed likely correlates to the ~44% of “normal” development embryos that the Lassy and Karr experiments observed. We thank the reviewer for bringing this to our attention. Our revisions (lines 559-572) include the addition of the Callaini reference as well as a more thorough discussion of the differences in experimental setups and potential reasons for why our results differ and are the same, as detailed above.

11. The phrase, (line 469) "completely separate from the first division defect" while clearly separate in time, should be mitigated to reflect the more reasonable phrase, "… more likely due to …" as the authors did above (line 458). Clearly, the authors have built a very strong case for these separate phenomena, but probably best to leave out the hyperbole.

We have rephrased this sentence to reflect the temporal distinction more clearly between the defects without hyperbole:

“The significant increase in mitotic errors observed in diploid CI-derived embryos relative to wild-type-derived embryos demonstrates the existence of a second, CI-induced defect, temporally and possibly mechanistically distinct from the first division defect” (lines 679-682).

12. Revise the abstract to acknowledge the previous literature that embryos can manifest with cytological errors hours into development, including in D. simulans. The authors communicated this situation nicely in the Introduction and can adopt the clarity in the abstract that the key question they intend to address is whether or not the chromosome errors at the 1st mitosis and later (both of which are well documented already) occur due to an initial error in the parental chromatin that carries forward or to independent new types of errors after the first mitosis. This is the question they aim to speak to in the work and it can be more clearly delineated as why it's important, what methods they used, and what the basic results were. For instance, this sentence could be revised from "In CI, embryos from crosses…fail to develop due to the immediate action of Wolbachia-produced factors in the first zygotic division" to "In CI, a paternally-delivered modification of the sperm leads to chromatin defects and lethality during and after the first mitosis of embryonic development in a variety of insect species" (published references: Culex, Aedes, Dmel, Dsim, etc).

We have revised the abstract as suggested:

“*Wolbachia*, a vertically transmitted endosymbiont infecting many insects, spreads rapidly through uninfected populations by a mechanism known as Cytoplasmic Incompatibility (CI). In CI, a paternally-delivered modification of the sperm leads to chromatin defects and lethality during and after the first mitosis of embryonic development in multiple species. However, whether CI-induced defects in later stage embryos are a consequence of the first division errors or caused by independent defects remains unresolved. To address this question, we focused on ~1/3 of embryos from CI crosses in *Drosophila simulans* that develop apparently normally through the first and subsequent pre-blastoderm divisions before exhibiting mitotic errors during the mid-blastula transition and gastrulation. We performed single embryo PCR and whole genome sequencing to find a large percentage of these developed CI-derived embryos bypass the first division defect. Using fluorescence in situ hybridization, we find increased chromosome segregation errors in gastrulating CI-derived embryos that had avoided the first division defect. Thus, *Wolbachia* action in the sperm induces developmentally deferred defects that are not a consequence of the first division errors. Like the immediate defect, the delayed defect is rescued through crosses to infected females. These studies inform current models on the molecular and cellular basis of CI” (lines 12-27).

13. In the intro, delete the word "sterility" used only in the first paragraph to unify the terminology with embryonic lethality or post-fertilization incompatibility in the paper since many readers may assume sterility refers to a pre-fertilization error that inhibits fertilization. That is not the case here.

As mentioned above, we have now revised this sentence:

“CI is a form of *Wolbachia*-induced paternal effect embryonic lethality” (line 40).

14. Line 113: It would help the paper and readers to change the subheader titles to emphasize the new discoveries of this study rather than what's known in previous literature. For instance, instead of stating CI produces late embryo lethality, add the new details of the percentage and type of delays seen here in the subhead.

We have revised the subheader titles as advised.

15. Line 121-122. D. simulans is also a useful model since it was previously shown to have defects after the first mitosis. That is worth noting at this point in the manuscript as it sets the stage for your study.

We thank the reviewer for this suggestion. We have revised this sentence to mention this with the appropriate references:

“We used *D. simulans* because CI is particularly pronounced in these species and results in defects both during and after the first zygotic mitosis” (lines 156-158).

16. Line 158: Cite Lassy and Karr 1996. -Lassy CW, Karr TL. Cytological analysis of fertilization and early embryonic development in incompatible crosses of *Drosophila* simulans. Mech Dev. 1996;57:47-58. -Callaini G, Riparbelli MG, Giordano R, Dallai R. Mitotic Defects Associated with Cytoplasmic Incompatibility in *Drosophila* simulans. Journal of Invertebrate Pathology. 1996;67:55-64.

We apologize for this omission. We have added these references.